# Thermo-Mechanical and Mechano-Thermal Effects in Liquids Explained by Means of the Dual Model of Liquids

Fabio Peluso 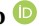

Leonardo SpA, Electronics Division, Via Monterusciello 75, 80078 Pozzuoli, Italy; fpeluso65@gmail.com

**Abstract:** We pursue to illustrate the capabilities of the Dual Model of Liquids (DML) showing that it may explain crossed effects notable in Non-Equilibrium Thermodynamics (NET). The aim of the paper is to demonstrate that the DML may correctly model the thermodiffusion, in particular getting formal expressions for positive and negative Soret coefficient, and another "unexpected" mechano-thermal effect recently discovered in liquids submitted to shear strain, for which the first-ever theoretical interpretation is provided. Both applications of the DML are supported by the comparison with experimental data. The phenomenology of liquids, either pure or mixtures, submitted to external force fields is characterized by coupled effects, for instance mechano-thermal and thermo-mechanical effects, depending on whether the application of a mechanical force field generates a coupled thermal effect in the liquid sample or vice-versa. Although these phenomena have been studied since their discoveries, dating back to the XIX century, no firm theoretical interpretation exists yet. Very recently the mesoscopic model of liquids DML has been proposed and its validity and applicability demonstrated in several cases. According to DML, liquids are arranged on a mesoscopic scale by means of aggregates of molecules, or *liquid particles*. These structures share the liquid world with a population of *lattice particles*, i.e., elastic waves that interact with the *liquid particles* by means of an inertial force, allowing the mutual exchange of energy and momentum between the two populations. The hit particle relaxes the acquired energy and momentum due to the interaction, giving them back to the system a step forward and a time-lapse later, alike in a tunnel effect.

**Keywords:** liquid modelling; phonons in liquids; Soret effect; thermodiffusion; heat propagation in liquids; shear waves in liquids; duality of liquids

---

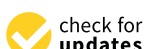



## 1. Introduction

Non-Equilibrium Thermodynamics (NET), also named Thermodynamics of Irreversible Processes, or again Thermodynamics of Phenomenological Processes, is a very robust method of physics to describe fluid systems, either gases or liquids (with appropriate restrictions also solids), subjected to boundary conditions which may vary vs. time and/or space [1]. NET is an extensive theory that follows from the Non-Equilibrium Statistical Mechanics, that in turn, other than those of the equilibrium theory, is based on the additional relevant postulate of the *time reversibility of the physical laws*. Such additional postulate states that "all the laws of physics remain unchanged upon time reversibility, i.e., if the time $t$ is replaced by $-t$, and if simultaneously the magnetic field $H$ is replaced by $-H$". We will meet again such a postulate when dealing with the DML in Sections 3 and 5. Typical phenomena described with the methods of NET are those in which transport processes take place, such as for instance the heat flow crossing a medium (described by the Fourier law), the mass flow due to a concentration gradient (Fick's law), the volume flow due to a pressure gradient (Poiseuille law), or also the electrical current due to a voltage difference applied to an electrical load (Ohm's law). The approach most largely adopted to describe such phenomenology in NET is due to Onsager [2,3], in which it is supposed that the gradients of some physical quantities, such as temperature, mass, pressure, voltage, etc, behave as *driving forces* (sometimes named *affinities*) producing *associated fluxes*, such as the

flux of heat, mass, volume, electrical charge, etc, similarly to the "cause-effect" relationship adopted in classical mechanics. This approach uses a mathematical matrix formalism and is based on four main hypotheses [1], namely:

a.   The quasi-equilibrium postulate, requiring that gradients (driving forces) are not too large.
b.   The linearity postulate, stating that all fluxes are linear functions of the relevant gradients (this is a consequence of the previous postulate, which allows a linear dependence of fluxes upon driving forces).
c.   Curie's postulate, constraining the tensor rank of coupled fluxes and forces.
d.   Onsager's reciprocal relations, requiring symmetric coefficient matrices in the force-flux relations.

Apart from the well-known phenomena listed above, there is a long list of "crossed" phenomena, in which a driving force is responsible also for other fluxes not strictly related to the applied force. Depending on which are the forces involved and the fluxes generated, such crossed effects are named, for instance, mechano-thermal, thermo-mechanical, thermo-electric, etc. Typical examples are the Soret, the Dufour, the Peltier, the Seebeck, the Thomson effects, etc. We will deal with in this paper mainly two of such coupled effects, one is the Soret effect [4,5] (with reference also to the Dufour effect), and the other is an "unexpected" mechano-thermal effect recently revealed in confined liquids under shear geometry [6–11].

The Ludwig–Soret effect, or simply Soret effect, is named after the German and Swiss researchers who in the XIX century independently observed that a temperature gradient applied to an isotropic mixture gave origin to a concentration gradient of the mixture components (other than the heat flow crossing the mixture). This phenomenon, generally named thermodiffusion, has been extensively experimentally investigated and many theoretical models have been proposed to interpret the experimental results, although none of them has provided yet a definitive explanation of all the characteristics of the Soret effect ([12–14] and references therein).

The "unexpected" mechano-thermal effect we will deal with has been discovered a few years ago by the group of Noirez and co-workers [6–11]. It is highlighted in two types of similar experiments, whose difference is the motion law of a movable disk, oscillating or following a Heaviside function. The mechano-thermal effect consists in both cases of the occurrence in a liquid in shear geometry of a temperature gradient due to the momentum transferred to it by a moving plate. Therefore, this mechano-thermal effect reveals a coupling between a mechanical force induced to the liquid by the shear strain which generates in turn the thermal gradient. Many interesting papers have been published showing, for instance, that the universal law $G' \approx L^{-3}$ for the low-frequency shear modulus $G'$ is confirmed also in confined liquids [15,16]; however, no firm theoretical explanation of such phenomenology at the mesoscopic level has been advanced yet.

In the frame of the liquid modeling at the mesoscopic scale, a new statistical model has been recently proposed, the Dual Model of Liquids (DML), whose validity and applicability have been shown in several cases [17–19] finding good correspondence with the experimental data for several liquid specific quantities, such as the thermal conductivity [17], the order-of-magnitude of the relaxation times governing the interaction process [17], the liquid specific heat [17,18]. It has also allowed providing a physical interpretation of the memory term appearing in the hyperbolic form of the heat propagation equation (Cattaneo) [19]. Shortly resuming the basis of the DML, this considers liquid molecules arranged at the mesoscopic level on metastable solid-like local lattices. Propagation of perturbations occurs at characteristic timescales typical of solids within these local domains of coherence, while mutual interactions of local clusters with inelastic wave-packets allow exchanging with them energy and momentum. The liquid is then assumed as a Dual System, the two subsystems being the *liquid particles* (i.e., the clusters of molecules) and the *lattice particles* (i.e., the wave-packets; we will use through the manuscript the terms *lattice particle*, wave-packet,

phonon, collective excitation on one side, or *liquid particle*, molecular cluster, and iceberg on the other, interchangeably, as synonyms).

The aim of this paper is to show that the crossed effects typical of NET may be fruitfully explained by means of the DML. The paper is organized as follows: in the second section the Soret and Dufour effects are shortly recalled as well as the main equations governing the phenomenon in NET. In the third section, the DML is briefly summarized, in particular, those aspects strictly related to the thermo-mechanical and mechano-thermal effects that are the topics of the present manuscript. It is then shown how the supposed duality of liquids in the DML characterized by the mechanism of "*lattice particle* $\leftrightarrow$ *liquid particle*" collision provides a physical explanation of the thermodiffusion as well as the sign dependence of the Soret coefficient $S_T$ upon several mixture' parameters. The fourth section is dedicated to the interpretation in the frame of the DML of the "unexpected" mechano-thermal effect detected in pure liquids. The agreement of the theoretical values obtained for the temperature drop with the experimental data supports the fact that the DML represents the first theory able to explain the conversion of mechanical energy into thermal energy in liquids. The Section 5 is aimed at discussing the implications of the results obtained in the previous sections. Finally, in the Section 6 other possible applications of the DML are anticipated.

## 2. The Thermodiffusion and the Soret Equilibrium in NET

NET is a phenomenological approach describing (mainly) fluid systems out of equilibrium. As such, it does not provide any physical interpretation of the phenomena under study, limiting its capabilities to describe them by means of macroscopic variables. When two or more processes occur simultaneously in a system, they may couple, generating very intriguing phenomena. Focusing our attention on phenomena involving only concentration and/or thermal gradients as generalized forces, Soret and Dufour effects are among the most known. The Soret effect is characteristic of the steady state that is reached in a fluid mixture, i.e., made by at least two chemical species, the solvent, and one solute, submitted to a temperature gradient. Even if such effects are known to occur also in multi-component mixtures, here we focus the attention only on two-components systems; the reasoning may be easily extended to multi-components mixtures, where the Soret coefficient $S_T$ becomes a vector. Figure 1 represents the typical situation one is dealing with in analyzing the thermodiffusion phenomenon. At the beginning the two chemical species are uniformly distributed; when the heat starts crossing the medium, from the hot to the cold side, one of the two species, for instance, the spheres of Figure 1, moves towards the cold side, while the other is pushed back towards the hot side (for the Newton' third law): this is the phenomenon of thermodiffusion, i.e., a mass flow produced by a heat flow. As soon as the thermodiffusion begins, as the spheres move along the heat current (and the cubes in the opposite direction), the back diffusion acts on the chemical species to balance the concentration gradient that is being established. Definitively, the displacement of the components of the solution generates at the steady state a concentration gradient for each of them, which in turn generates two mass flows of the chemical species, coupled with the heat flux due to the external temperature gradient. At the steady-state, or the Soret equilibrium, the thermodiffusion is balanced by the ordinary diffusion; one may observe a temperature gradient (external constraint) responsible for a heat flow, and two opposite concentration gradients (crossed effect), for the spheres and the cubes. Generally, the attention is focused on the less concentrated component of the mixture, the solute. Such dynamical equilibrium, the Soret, is described by means of the Soret coefficient $S_T$, which is positive when the solute drifts towards the cold side, while it is negative when it drifts towards the hot one. The Dufour effect is the opposite of the Soret. When there is an initial gradient of concentration in a fluid mixture, i.e., a gradient of chemical potential, it is known that the system spontaneously evolves towards the equilibrium due to the ordinary Fick diffusion. A careful evaluation of the thermal evolution of the system shows that the spontaneous

diffusion of the chemical species of a mixture to establish a uniform distribution generates a temperature gradient: this phenomenon is named the Dufour effect in NET.

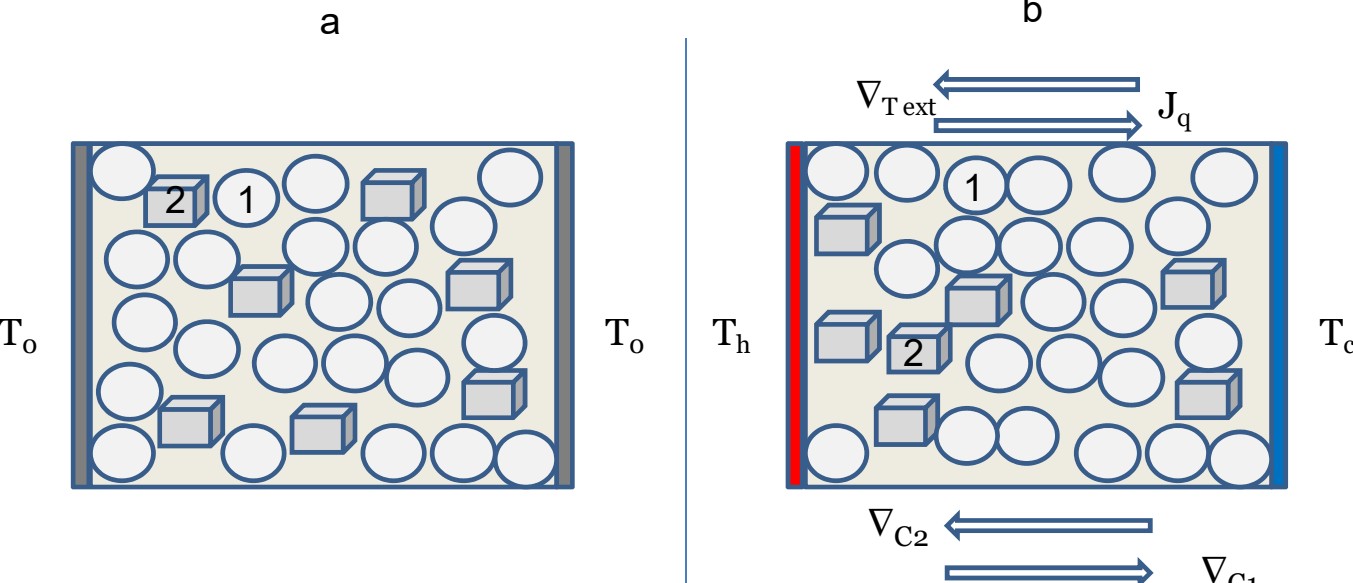

**Figure 1.** Schematic representation of a mixture made by two different chemical species, 1 and 2, ideally represented by spheres and cubes. The solids represent the icebergs that in the DML constitute the liquid matrix. A temperature gradient $\nabla T_{ext}$ is applied to the medium generating a heat flux $J_q$. At the beginning of the experiment (**a**) the two species are uniformly distributed. As far as the heat crosses the medium, the two species separate, until a steady state is reached, characterized by the separation of the two chemical species (**b**). The steady state is the Soret equilibrium, characterized by the dynamical equilibrium reached by the two diffusive mechanisms, one driven by the temperature gradient, the other by the concentration gradients of the chemical species. The arrows represent the direction of the related vectors indicated at their tips.

The Soret equilibrium is characterized by a very small concentration gradient, a second-order effect, induced by the heat flow. Nevertheless, because coupled heat and mass fluxes occur in many relevant processes, such as distillation, extraction, crystallization, absorption, drying, or even condensation, they can play an important role in many natural [20–32] and industrial processes [33]. Thermodiffusion for instance may reveal useful to separate mixtures that are hardly handled by conventional methods [34], such as isotope fractionation [35], colloids separation [36], surface coating [37], enriched combustion [33], etc. We will deeply analyze in this paper the thermodiffusion and the Soret equilibrium, although the reader may certainly understand how our reasoning may be easily applied to other cross phenomena typical of NET. The scientific literature on thermodiffusion is very wide, encompassing the experimental investigation [38–44] and theoretical modeling [45–64] developed in the past two centuries after its discovery. Because this effect is typical of fluid mixtures kept in thermal non-equilibrium, it is very difficult to get the experimental setup sufficiently stable to avoid convection due to the coupling of the earth's gravity with the thermal field. For such reason, the thermodiffusion and the Soret equilibrium have been studied for 20 years or more in microgravity environments [65–77], exploiting the virtual absence of gravity typical of platforms orbiting around the earth. In the literature a by far non-exhaustive list of references is reported, to which the reader may refer; consequently we do not want to enter here into the historical developments of the topic, but we will refer to some of the most important references to point out the capabilities of the DML with respect to previous models. For the sake of clarity, we report a brief description of the governing equations for thermodiffusion.

Considering temperature and concentration gradients as generalized forces and heat and mass current as generalized fluxes, the set of phenomenological equations in NET needed to describe the observed phenomena in a 2-components fluid system are [1–5]:

$$J_q = -L_{qq} \frac{\nabla T}{T^2} - L_{q1} \left( \frac{\partial \mu_1}{\partial w_1} \right)_{T,p} \frac{\nabla w_1}{(1 - w_1)T} \tag{1}$$

$$J_m^1 = -L_{1q} \frac{\nabla T}{T^2} - L_{11} \left( \frac{\partial \mu_1}{\partial w_1} \right)_{T,p} \frac{\nabla w_1}{(1 - w_1)T} \tag{2}$$

where $J_q$ and $J_m^1$ are the heat and mass fluxes, $T$ the temperature, $p$ the pressure, $\mu_1$ and $w_1$ the chemical potential and mass fraction of component 1 (the solute for instance), $L_{qq}$, $L_{1q}$, $L_{q1}$, $L_{11}$ the phenomenological Onsager coefficients describing the proportionality between the generalized forces and the fluxes they generate. In particular, $L_{1q}$ and $L_{q1}$ are the cross-correlated phenomenological coefficients, for which $L_{1q} = L_{q1}$ holds in the Onsager framework (reciprocity postulate). Equations (1) and (2) are often re-written in terms of experimental coefficients instead of Onsager's ones, in the following form (constitutive equations):

$$J_q = -K \nabla T - \rho w_1 \left( \frac{\partial \mu_1}{\partial w_1} \right)_{T,p} T D_T^D \nabla w_1 \tag{3}$$

$$J_m^1 = -\rho w_1 w_2 D_T^S \nabla T - \rho D \nabla w_1 \tag{4}$$

where $K$ is the thermal conductivity of the liquid mixture, $\rho$ its mass density, and $D$, $D_T^S$ and $D_T^D$ the ordinary Fick diffusion coefficient, the Soret and the Dufour thermal diffusion coefficients, respectively. The Soret equilibrium is attained at the steady state, when the heat still flows through the mixture but there is no more mass diffusion so that $J_m^1 = 0$ in Equation (4); eventually we get:

$$S_T = \frac{D_T^S}{D} = -\frac{1}{w_1 w_2} \left( \frac{\nabla w_1}{\nabla T} \right)_{J_m^1 = 0} \tag{5}$$

which defines the Soret coefficient $S_T$ and makes it evident that the equilibrium is attained by balancing the thermodiffusion and the ordinary diffusion. Comparing Equations (1)–(4), one may easily get the following expressions for the two thermal diffusion coefficients:

$$D_T^S = \frac{L_{1q}}{\rho w_1 w_2 T^2} \tag{6}$$

$$D_T^D = \frac{L_{q1}}{\rho w_1 w_2 T^2} \tag{7}$$

From the Onsager's reciprocity relations, it follows that $D_T^S = D_T^D$. Finally, the Fick coefficient of ordinary diffusion $D$ is related to $L_{11}$ through:

$$D = L_{11} \frac{1}{\rho w_2 T} \left( \frac{\partial \mu_1}{\partial w_1} \right)_{T,p} \tag{8}$$

The Heats of Transport are other parameters often used in NET to describe the Soret or Dufour effects. We will take care of them elsewhere.

## 3. The Thermodiffusion and the Soret Equilibrium in the DML

In the first part of this section, we will shortly gather the main and relevant aspects of the DML avoiding repeating the concepts introduced and extensively discussed elsewhere. Readers interested in the details of the model are invited to refer to the literature [17–19]

where the theoretical developments, numerical simulations, experimental results, and applications are collected and discussed in depth. The second part is dedicated to showing that the DML may provide a completely new approach to the physical interpretation of the thermodiffusion phenomenon and of the Soret equilibrium occurring in liquid mixtures to which a temperature gradient is applied (when the phenomenon involves macromolecular solutes, it is generally named thermophoresis). A similar approach can be applied of course also to the Dufour effect, as well as to other phenomena typical of NET. In particular, we will show how the DML is able to model the thermodiffusion, taking into account even the sign variation that sometimes occurs for the same couple of solute-solvent of a mixture at varying parameters, such as the average temperature, the intensity of the temperature gradient or also the initial concentration. The capability of the DML to justify that the third law of dynamics is not falsified in such type of experiments, as some authors have supposed ([12–14] and references therein), is dealt with in the Section 5.

Two facts make challenging the experimental determination of the Soret coefficient, namely its small magnitude, usually of the order of $(10^{-4} \div 10^{-3})\,°C^{-1}$, and the presence of undesirable convective disturbances due to the coupling of the temperature gradient with gravity, sometimes even present in microgravity environments [66,68–77]. It is then not surprising that the several theoretical models proposed through the years [12,13] often fail when dealing with peculiar characteristics of $S_T$, such as its sign inversion or the presence of minima. Experimental [38–46], theoretical [36,45–64] and computational [12–14] studies on thermodiffusion and Soret equilibrium have significantly advanced in recent years, but several intriguing and fundamental questions still remain unanswered. One such question, maybe the most relevant, is the origin of the forces driving the observed phenomenology at the micro-mesoscopic scale. We will try to fill this gap by means of DML.

The DML rests on two hypotheses, both with an experimental background, namely: (1) experiments performed by means of Inelastic X-ray Scattering (IXS) and Inelastic Neutron Scattering (INS) techniques [78–97] allowed to discover that the mesoscopic structure of liquids is characterized by the ubiquitous presence of solid-like structures, whose size is of a few molecular diameters, through which elastic waves propagate as in the corresponding solid phase; the number and the size of these structures depend upon the temperature and pressure of the medium. (2) Elastic energy and momentum in liquids propagate by means of collective oscillations, or wave-packets, similarly to phonons in crystalline solids (this hypothesis is a consequence of the first). Although both arguments have been largely and deeply dealt with in [17] where DML is introduced, it is the case to recall the attention of the reader on the relevance of the INS and IXS experiments findings, whose results are at the basis of a dual vision of the liquid modeling at the mesoscopic scale. In fact, even if the frequency ranges of these experiments are high, typically from less than 1 to 2–3 THz, however, the fact that the propagation speed of elastic waves in the pseudo-crystalline structures of liquids are higher than those typical of liquids, and closer to those of solids, ensures that these speeds are maintained over relatively long distances, covering several molecular diameters, i.e., the dimensions of the clusters (see also the Section 6 for further recent findings on this topic). In fact, the experimental results show that, once these distances are exceeded, the propagation speeds of the elastic waves return to the range of values typical of liquids, thus providing a signature of the borders of the pseudo-crystalline structures. Among the many experimental works performed on such topic, other than that of Ruocco et al. [81] who measured the speed of sound in water amounting to $\approx 3200\,m/s$ on length scale of few molecular diameters, it is the case of citing that of Giordano and Monaco [97]. They performed the first measurements across the glass-liquid and solid-liquid transitions and reported on a comparison of the collective excitations in liquid and polycrystalline sodium. As it concerns liquid sodium, it exhibits acoustic excitations of both longitudinal and transverse polarization at frequencies strictly related to those of the corresponding crystal. The relevant difference between the liquid and the crystal is the broadening of the excitations in the case of liquid because of an additional disorder-induced contribution coming into play. The authors deduce a direct connection

between the structural and dynamic properties of liquids, with short-range order and over-all structural disorder characterized by specific fingerprints. To confirm the validity of the DML, we evaluated the OoM of the phonon mean free path $\langle \Lambda_0 \rangle$ and lifetime $\langle \tau_0 \rangle$ within a *liquid particle* [17]. With reference to the experimental values, $\langle \Lambda_0 \rangle$ will be a multiple of the phonon wavelength $\lambda^0$, $\langle \Lambda_0 \rangle = n\lambda^0$, and $\langle \tau_0 \rangle$ of $\tau = 1/\nu^0$, $\langle \tau_0 \rangle = n/\nu^0$, with $n > 1$. Using the data for the water of [79,94], typical values for the parameters characterizing a phonon (variation range is function of temperature, pressure, and $q$ orientation) are: (central) frequency $\frac{n}{\langle \tau_0 \rangle} = \langle \nu^0 \rangle \approx 0.95 \div 2.5$ THz, wave-length $\frac{\langle \Lambda_0 \rangle}{n} = \langle \lambda^0 \rangle \approx 1 \div 3$ nm and velocity $\frac{\langle \Lambda_0 \rangle}{\langle \tau_0 \rangle} = \langle \lambda^0 \rangle \cdot \langle \nu^0 \rangle = \langle u^0 \rangle \approx 3.1 \div 3.4 \cdot 10^3$ m/s. Interestingly, this value fits very well with the experimental data obtained for the propagation velocity of thermal waves in water [81]. The value of $\langle \Lambda_0 \rangle$ represents also the typical OoM of the size of a *liquid particle* at the exchanged momentum $q$, allowing us to confidentially assume a value of few units for $n$. A theoretical explanation of the occurrence of a solid-like value of sound propagation in liquids is based on the points of view of Brillouin and Frenkel. We may distinguish two limit regimes for the propagation of perturbations in liquids. One is the "normal" hydrodynamic regime, in which one considers the medium as a continuum and the excitations on a so long timescale that the system may be assumed to be in thermo-dynamic equilibrium. This regime is also usually referred to as viscous. The second is the "solid-like" or elastic regime, in which the dynamics becomes that of a free particle between successive elastic collisions. The investigation of collective dynamics becomes of particular interest when intermediate time- and length-scale are considered, i.e., distances comparable to those characterizing the structural correlations among particles, and times comparable to the lifetimes of such correlations. Although these two extreme behaviors, viscous and elastic, are well known in physics, the intermediate situation is still far from being fully characterized. This intermediate range is referred to as viscoelastic regime. It is just in this regime that the DML is applied, as is clear also from the characteristics of the elementary interactions described in the following.

According to this view, thermal energy is considered a form of elastic energy, as supposed by Debye [98,99], Brillouin [100,101], and Frenkel [102]. At the mesoscopic level the liquid phase is modeled in the DML as constituted by two sub-systems, mutually interacting among them, the *liquid particles*, i.e., a sort of solid-like aggregates of liquid molecules, and the wave-packets, or *lattice particles*. Therefore we are led to define a "liquid" in the DML not as a liquid in the classical assumption, but as a mixture of solid icebergs (the solid-like part) and of an amorphous phase (the classical liquid part). It will not have escaped a very attentive and informed reader that this approach is not completely new in the world of the physics of liquids. First, it recalls that of Brillouin [17,100,101] and Frenkel [102]; Landau also described the HeII as made of "normal" and "superfluid" parts [103]. Eyring in his Reaction Rate Theory [104–106], also known as the "hole" theory, had supposed that a liquid could be assumed to have a quasi-crystalline structure. Similarly to a gas, the liquid should consist of molecules moving around empty spaces, or holes. A molecule is pictured as vibrating around an equilibrium position until it acquires sufficient energy to overtake the attracting forces "and" a hole is available in the nearest neighbor where the molecule can jump (see also Figure 2 in [17]). More recently, Andreev [107] in a very interesting paper assumed that a liquid consists of two weakly coupled systems, the phonons and the remainder of the liquid. He calculates the effects connected with the presence of weakly damped phonons in a normal liquid. In particular, he calculated the expressions for mechano-thermal and thermo-mechanical effects, as well as the propagation of shear oscillations (see also the Section 6 for recent experimental findings on this topic).

Figure 2 shows in a pictorial way the model we have in mind. Solid-like icebergs, the *liquid particles*, are dispersed in the amorphous phase of liquid, while thermo-elastic waves, the *lattice particles*, travel through them. As far as the perturbations travel inside a solid-like structure, they show a solid-like character, i.e., they are (quasi) harmonic waves traveling at speeds close to that of the corresponding solid phase (3200 m/s in the case of water, [17,18,81]). When however they leave the *liquid particle* and travel through

the amorphous environment, lose the harmonic outfit and become wave-packets. The interaction itself between these waves and the solid-like icebergs has an anharmonic character, allowing momentum and energy to be exchanged between the two reservoirs, the *liquid particles* and the *lattice particles*. The momentum transferred $\Delta p^{wp}$ generates the force $f^{th}$

$$f^{th} = \frac{\Delta p^{wp}}{\langle \tau_p \rangle} = -\nabla \phi^{th} \tag{9}$$

responsible for the energy and mass diffusion in liquids [17–19]; $\langle \tau_p \rangle$ represents the (finite) duration of the *lattice particle* $\leftrightarrow$ *liquid particle* interaction and $\phi^{th}$ is the anharmonic interaction potential between the *lattice particle* and the *liquid particle*, given by [17,19] (for a discussion about the functional dependence of $\phi^{th}$ upon distance and about its quantum/classical form, the reader may refer to the Discussion in [17]):

$$-\nabla \phi^{th} = f^{th} = \sigma_p \delta \left( \frac{J_q^{wp}}{u_\varphi} \right) \tag{10}$$

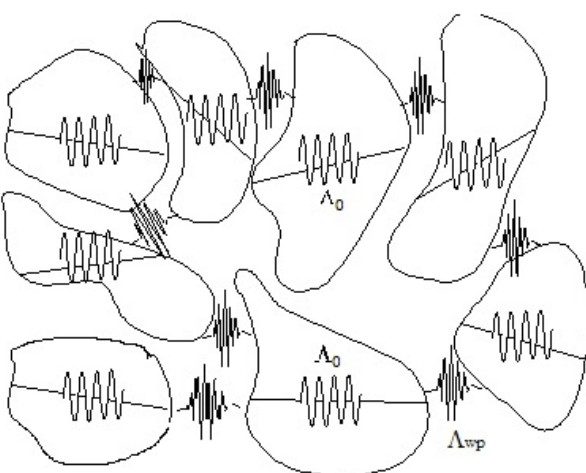

**Figure 2.** Icebergs of solid lattice fluctuating and interacting within the liquid global system at equilibrium. As far as elastic (thermal) perturbations propagate within an iceberg, they behave as in solids. Propagation velocity has then the values typical as those of the solid lattice, as found by Ruocco et al. [81] of about 3200 m/s for the case of water. Average sizes of icebergs $\langle \Lambda_0 \rangle$ have been estimated of the order of magnitude of some nanometers. When perturbations cross the boundary between two icebergs, $f^{th}$ develops and energy and momentum are transmitted from one to the nearest-neighbor iceberg. This pictorial model of liquids at mesoscopic scale, on which the DML is based, reflects also what may be deduced from experiments performed with IXS techniques, able to observe liquids at such scale-lengths. In a solution, solute particles may be considered as icebergs having elastic impedance different from that of the solvent. Energy and momentum exchanged between the two types of icebergs produce a net effect resulting in the diffusion of the solute along the concentration gradient. If a temperature gradient is imposed externally, the net effect will depend on the prevailing flux of wave packets, which will give rise to thermal diffusion of one species with respect to the other.

Here $u_\varphi$ is the phase velocity associated with the wave-packet and $\sigma_p$ is the cross-section of the "obstacle", the solid-like cluster, on the surface of which $f^{th}$ is applied; $J_q^{wp}$ the energy flux carried by wave-packets [17]. Some considerations on $f^{th}$, or $\phi^{th}$, are in order. First, $\phi^{th}$ is supposed to be anharmonic because of the inelastic character of the interaction, which is not instantaneous but lasts $\langle \tau_p \rangle$ during which the particle is displaced by $\langle \Lambda_p \rangle$ (here and in the rest of the paper, the two brackets $\langle \rangle$ indicate the average over a statistical ensemble of the quantity inside them). Second, $f^{th}$ can either be positive or negative

depending on whether the quantity $\left( J_q^{wp} / u_\phi \right)$ increases or decreases as a consequence of the interaction. This point is connected with the last equality in Equation (10) which deserves a more accurate analysis because it is pivotal for the purposes of the present manuscript. It is indeed well known that a wave impinging on an obstacle exerts on it a pressure. The radiation pressure was introduced for the first time by Balfour Stewart in 1871 [20] for the case of thermal radiation. Maxwell took up this theme in Arts. 792–793 of his Treatise in 1873 [21], arguing on the basis of his stress tensor. The first convincing experimental evidence for the radiation pressure of light was given by Lebedev in 1901 [22] (the Crookes radiometer does not demonstrate electromagnetic radiation pressure). We have shown [17] by means of the radiant vector $\vec{R}$

$$\vec{R} = \left( \Delta \Pi \cdot \dot{\xi} \right) \vec{r} = \vec{J_{el}} \tag{11}$$

that this concept may be generalized and used for any type of wave ($\dot{\xi}$ is the oscillation velocity of the particles, $\vec{r}$ the unit vector along the direction of propagation and $\vec{J_{el}}$ the flux of elastic energy generating the pressure $\Pi$). Here we exploit again this concept and apply it to the momentum transferred by elastic, or thermal, waves that impinge on an obstacle, such as for instance a solute molecule in a liquid mixture. Accordingly, we replace the gradient of elastic pressure with that of thermal energy, yielding:

$$\nabla \Pi^{th} = \nabla \left( \frac{J_q^{wp}}{u_\varphi} \right) = -\psi^{th} \tag{12}$$

Equation (12) contains two supplementary information; first, because thermal energy is carried by elastic wave-packets, the velocity of energy propagation remains the same. Second, the gradient of pressure gives rise to an inertial term, $\psi^{th}$, dimensionally a force per unit of volume, responsible for momentum transport associated with the wave-packets propagation. Equation (12) is valid when the characteristics of the medium change continuously along the wave-packets propagation direction. The same argument may be used when wave-packets impinge on an "obstacle", as for instance a *liquid particle*, either of solute or of solvent: the radiant vector changes and a thermal radiation pressure $\Pi^{th}$ is produced on the boundary. Observing that the inelastic character of the collision allows the exchange of momentum, this leads to the appearance of the force $f^{th}$, acting on molecular clusters. By rewriting Equation (12) in terms of discrete quantities, one yields the following expression for the net $\Pi^{th}$:

$$\Delta \Pi_{th} = \left[ \left( \frac{J_q^{wp}}{u_\varphi} \right)_1 - \left( \frac{J_q^{wp}}{u_\varphi} \right)_2 \right] = \frac{f^{th}}{\sigma_p} \qquad \Rightarrow \tag{13}$$

$$f^{th} = \sigma_p \left[ \left( \frac{J_q^{wp}}{u_\varphi} \right)_1 - \left( \frac{J_q^{wp}}{u_\varphi} \right)_2 \right] \tag{14}$$

We conclude that the last equality in Equation (10) provides the algebraic sign for $f^{th}$ according to the sign of the quantity in square brackets of Equation (14), i.e., on whether the energy associated with the propagation of elastic waves in the medium "1" is larger or smaller than that in the medium "2". Expressions analogous to Equation (13) or Equation (14) have been previously derived by many authors following different approaches [17,20–26]. It derives from the Boltzmann–Ehrenfest' Adiabatic Theorem [27], later generalized by Smith [28] and by Gaeta et al. [29]; another approach gave analogous results by calculating the radiation pressure produced by acoustic waves on an idealized liquid-liquid interface [30].

Let's now go a little bit more into detail on the characteristics of $f^{th}$ in the DML, i.e., on its intensity, orientation, functional dependence, etc. Everything begins by remembering that $q_T$

$$q_T = \int_0^T \rho C_V d\theta = f[\Theta/T] \tag{15}$$

is the internal energy per unit of volume of a liquid at temperature $T$, $\rho$ the medium density, $C_V$ the specific heat at constant volume per unit mass, $\Theta$ the Debye temperature of the liquid at temperature $T$. Since this dynamics occurs mainly at high frequencies and involves only the DoF of the lattice, we introduce the parameter $m$ to account for the average number of DoF participating in the collective dynamics. Defining $m$ as the ratio between the number of collective DoF surviving at temperature $T$ and the total number of available collective DoF, the fraction $q_T^{wp}$ of $q_T$:

$$q_T^{wp} = m q_T = \langle \mathcal{N}^{wp} \cdot \varepsilon^{wp} \rangle = m \frac{\int_0^T \rho C_V d\theta}{\rho C_V T} \rho C_l T = m^* \rho C_l T \tag{16}$$

accounts for that part of the thermal energy transported by the wave-packets (the definition of $m*$ is easily deduced). Of course, it holds $0 \leq m \leq 1$ [17]. $\mathcal{N}^{wp}$ is the number of wave-packets per unit of volume and $\varepsilon^{wp}$ their average energy. $q_T^{wp}$ is propagated through the liquid by means of inelastic interactions between the *liquid particles* and the *lattice particles*, i.e., the wave-packets that transport the thermal or, generally, the elastic energy. Figure 3 exemplifies the elementary physical mechanism behind the DML. The *lattice particle* $\leftrightarrow$ *liquid particle* interaction allows the exchange of both energy and momentum between the phonon and the *liquid particle*. To make more intuitive the model, we have ideally divided the elementary *liquid particle* $\leftrightarrow$ *lattice particle* interaction into two parts: one in which the *lattice particle* collides with the *liquid particle* and transfers to it momentum and energy (both kinetic and potential) and the other in which the *liquid particle* relaxes returning the energy to the thermal reservoir through a *lattice particle*, alike in a tunnel effect. Considering an event of type (a) of Figure 3, an energetic wave-packet collides with a *liquid particle* that acquires the momentum $\Delta p^{wp}$ (Equation (16)) and the energy $\Delta \varepsilon^{wp}$:

$$\Delta \varepsilon^{wp} = h\langle \nu_1 \rangle - h\langle \nu_2 \rangle = \Delta E_p^k + \Delta \Psi_p \tag{17}$$

In Equation (17) $\langle \nu_1 \rangle$ and $\langle \nu_2 \rangle$ represent the wave-packet (central) frequency [17] before and after the collision, respectively. Part of $\Delta \varepsilon^{wp}$ becomes the kinetic energy $\Delta E_p^k$ acquired by the *liquid particle*, and the remaining part becomes $\Delta \Psi_p$, i.e., the potential energy of internal DoF of the solid-like cluster. As a consequence of the collision (a), the wave-packet loses energy and momentum, which are acquired by the cluster and converted into kinetic and potential energy. The kinetic energy is dissipated as friction against the liquid. The potential energy $\Delta \Psi_p$ is relaxed by the *liquid particle* once the interaction is completed, its dissipation lasts $\langle \tau_R \rangle$ during which the *liquid particle* travels by $\langle \Lambda_R \rangle$, at the end of which the residual energy stored into internal DoF returns to the liquid reservoir. The total displacement of the particle, $\langle \Lambda \rangle$, and the total duration of the process, $\langle \tau \rangle$, are:

$$\langle \Lambda \rangle = \langle \Lambda_p \rangle + \langle \Lambda_R \rangle \tag{18}$$

$$\langle \tau \rangle = \langle \tau_p \rangle + \langle \tau_R \rangle \tag{19}$$

There are several interesting observations at this point that we want to propose to the reader's attention. First, as pointed out before, the elementary interaction looks like a tunnel effect, inasmuch as the energy subtracted from the phonon's reservoir returns to it a time interval $\langle \tau \rangle$ later and a step $\langle \Lambda \rangle$ forward. We want to clarify however that the purely classical "tunnel effect" occurring in the *liquid particle* $\leftrightarrow$ *lattice particle* interaction

should not be confused with tunneling in physics, which is ordinarily associated with quantum phenomenology. We have adopted such a terminology only to give the reader the idea of what happens in a liquid upon the interaction, but there are no hidden quantum phenomena beyond such a word. What matters here to the scope of DML is the dynamics linked to the relaxation time, i.e., to recognize that $\langle \tau \rangle$ is the time interval during which the energy disappears from the liquid thermal reservoir because it is trapped in the internal DoF; once $\langle \tau \rangle$ has elapsed, it reappears in a different place. In this way, relaxation times, introduced ad hoc by Frenkel [102], find a clear physical interpretation (interestingly, $\langle \Lambda \rangle$ and $\langle \tau \rangle$ in Equations (18) and (19) have the same meaning as $\delta$ and $\tau_F$ defined by Frenkel [102], see also Figure 2 in [17]). The second point recalls us to the mind that in this manuscript we are dealing with the Soret equilibrium, i.e., a phenomenon characterized by a dynamical equilibrium between the diffusion of particles due to the heat crossing the mixture, that is balanced by the Fick back diffusion due to the concentration gradient that is being established by the thermodiffusion. In the papers published so far [17–19], we have mainly been concerned with that part of the energy lost by the wave-packets and converted into internal (potential) energy of *liquid particles*, $\Delta \Psi_p$, and with the consequences of its propagation. At equilibrium, the variation of potential energy $\Delta \Psi_p$ is zero in the average because the thermal content of the system does not vary anymore, as well as the number of DoF excited. Accordingly, the only term surviving in the third member of Equation (17) is that accounting for the average kinetic energy acquired by the *liquid particle*, $\Delta E_p^k$. Apart from this, in the frame of DML, we are instead concerned with a flux of momentum, in fact, we are analyzing the mechanism responsible for the solute, or solvent, molecules flux pushed by a flux of thermal (elastic) energy crossing the liquid and due to an external source, the temperature gradient applied to the system (a liquid mixture). We are then concerned here with that part of energy lost by wave-packets and converted into kinetic energy of liquid particles.

We want now to draw the attention of the reader to the event of type (b) of Figure 3 which is the time-reversal of (a): an energetic cluster interacts with a wave-packet transferring to it energy and momentum. The outgoing wave-packet has momentum and energy larger than the incoming one so that the net effect of the interaction is to increase the liquid thermal energy carried by phonons at the expense of the kinetic and internal energy of the liquid particle. Another crucial point relevant to our purposes is that at (thermal or concentration) equilibrium conditions, events (a) will alternate over time with events (b), to keep statistically equivalent the balance of the two energy reservoirs. The effect of the interactions is null if integrated over the entire surface of the particle since its thermal regime does not change, nor does it move; consequently the mesoscopic equilibrium induces the macroscopic equilibrium, following the Onsager postulate [1–3,17]: events like those of Figure 3 will be equally probable along any direction, with a null average over time and space. On the contrary, when the symmetry is broken by an external force field, for instance by a temperature (Soret) or a concentration (Dufour) gradient, type (a) events will prevail over type (b), or vice-versa, along the direction of the gradient; the gradient will generate an imbalance of the collisions and possibly also a variation of the energy of the phonons (their number does not change as a first approximation) and an imbalance of the number of collisions. We will return to this point when discussing the non-equilibrium phenomena which are the topic of the present manuscript.

Figure 4 represents a close-up of Figure 3a and shows how the energy $\Delta \Psi_p$ is propagated inside a *liquid particle* until it is given back to the thermal reservoir. The interaction begins with a wave-packet impinging on a *liquid particle*, transferring to it energy and momentum. $\langle \Lambda_{wp} \rangle$ is the extension of the wave-packet and $\langle d_p \rangle$ that of the liquid particle. Because the *liquid particle* in the DML is considered a solid-like structure, following the experimental evidences obtained by means of IXS and INS experiments [17,78–97] and the pioneering intuitions of Brillouin [101] and Frenkel [102], the phonon's flight through the *liquid particle* behaves like a (quasi-) harmonic wave, crossing it at a speed close to that of the corresponding solid phase. Once $\langle \tau_p \rangle$ has elapsed, the *liquid particle* has travelled by

$\left\langle \Lambda_p \right\rangle$, it relaxes the energy stored in the internal DoF, then it travels again by $\left\langle \Lambda_R \right\rangle$ during $\left\langle \tau_R \right\rangle$ (not shown in the figure). The presence of a relaxation time is a signature of liquids allowing to justify, among others, some of their peculiar characteristics [17,19].

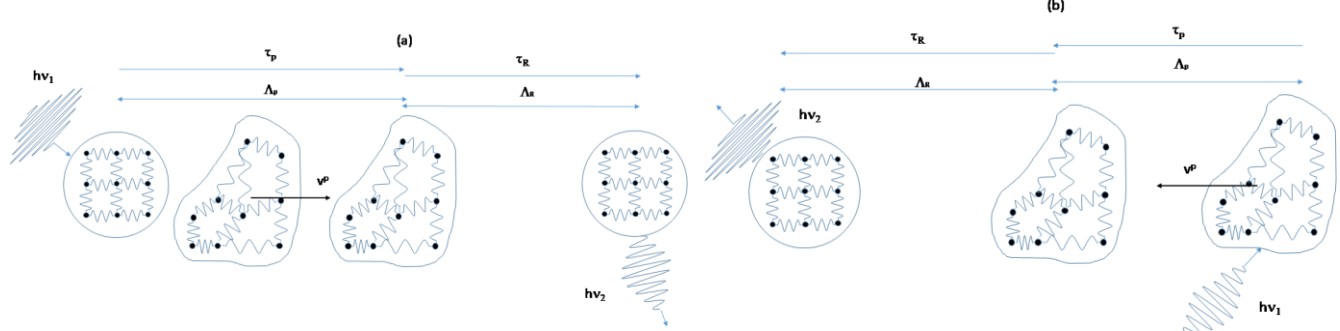

$$\Delta \varepsilon^{wp} = h\left\langle \nu_1 \right\rangle - h\left\langle \nu_2 \right\rangle = \Delta E_k^p + \Delta \Psi^p$$
$$\Delta p^{wp} = f^{th} \cdot \left\langle \tau_p \right\rangle$$

**Figure 3.** Schematic representation of inelastic collisions between wave-packets and liquid particles. The dots represent the molecules, arranged in a metastable *liquid particle*, the small springs indicate the forces involving the internal DoF and responsible for the temporary stability of the cluster. In the event represented in (**a**), an energetic wave-packet transfers energy and momentum to a liquid particle; it is commuted upon time reversal into the one represented in (**b**), where a liquid particle transfers energy and momentum to a wave-packet. The particle changes velocity and the frequency of wave-packet is shifted by the amount $(\nu_2 - \nu_1)$. Due to its time symmetry, this mechanism has been assumed the equivalent of Onsager's reciprocity law at microscopic level [17,19]. This elementary interaction has an activation threshold for potential energy and, depending on how much energy is absorbed by internal DoF, the rest will become kinetic energy of the liquid particle. When a system is subjected to an external temperature gradient, the first effect that occurs is the establishment of the internal temperature gradient, according to the Cattaneo–Fourier equation. Once the internal DoF have reached the statistical equilibrium in terms of distribution of their degree of excitation depending on the local temperature, the phenomenon of matter diffusion develops due to the temperature gradient and at the expense of the kinetic energy reservoir. In other words, inelastic effects dominate the dynamics during the transient phase, leaving the control to the kinetic reservoir at the steady-state, where only the elastic effects of the elementary interactions matter.

A question that the reader is certainly wondering at this point is "what pushes and drives the wave-packets in liquids between two successive collisions?". To answer such question we should take in mind that in DML the wave-packet current is always present in liquid, even in isothermal conditions. During their free-flight between two successive interactions (Figure 2) with the *liquid particle*, the wave-packets are considered in the DML as local microscopic heat currents. Each *lattice particle* propagates along an average distance $\left\langle \Lambda_{wp} \right\rangle$, during a time interval $\left\langle \tau_{wp} \right\rangle$, which is its average life-time. Accordingly, the ratio $\left\langle \Lambda_{wp} \right\rangle / \left\langle \tau_{wp} \right\rangle$ defines the wave-packet average propagation velocity between two successive interactions, $u^{wp} = \dfrac{\left\langle \Lambda_{wp} \right\rangle}{\left\langle \tau_{wp} \right\rangle} = \lambda^{wp} \cdot \nu^{wp}$, where $\nu^{wp}$ and $\lambda^{wp}$ are the (central) frequency and wave-length of the wave-packet (although an elastic perturbation of the liquid lattice should be represented by a wave-packet, whose localization depends on the microscopic structure to which it is associated, here it will be represented, for the sake of mathematical simplification, before and after an interaction, as a monochromatic wave, longitudinally polarized, and hence identified by its average wave-length $\lambda^{wp}$ and frequency $\nu^{wp}$, whose product is equal to $u^{wp}$). Considering a surface S arbitrarily oriented within the liquid, at thermal equilibrium an equal number of wave-packets will

flow through it in opposite directions. Let's assume now that locally every heat current, although isotropically distributed in the liquid, is driven by a *virtual temperature gradient* $\left\langle \frac{\delta T}{\delta z} \right\rangle$ applied over $\langle \Lambda_{wp} \rangle$. From Equation (16), the variation of the energy density within the liquid along the "+z" direction is:

$$\delta_{+z} q_T^{wp} = \frac{1}{6} \delta(m q_T) = \frac{1}{6} \frac{\partial q_T^{wp}}{\partial T} \delta T = \frac{1}{6} \frac{\partial q_T^{wp}}{\partial T} \left\langle \frac{\delta T}{\delta z} \right\rangle_{+z} \langle \Lambda_{wp} \rangle \tag{20}$$

where the local virtual temperature gradient $\left\langle \frac{\delta T}{\delta z} \right\rangle$ in the last member represents the thermodynamic "*generalized*" force (with the meaning of NET) driving the diffusion of thermal excitations along *z*. The factor 1/6 accounts for the isotropy of the phonon current through a liquid at equilibrium. By applying this driving force over the distance $\langle \Lambda_{wp} \rangle$, a wave-packet diffusion, i.e., the heat current $j^{wp}$, is generated along the "+z" direction:

$$j_{+z}^{wp} = -D_+^{wp} \left( \frac{\delta q_T^{wp}}{\delta z} \right)_{+z} = -\frac{1}{6} u^{wp} \langle \Lambda_{wp} \rangle \frac{\partial q_T^{wp}}{\partial T} \left\langle \frac{\delta T}{\delta z} \right\rangle_{+z} \tag{21}$$

where $D_+^{wp} = \frac{1}{6} u^{wp} \langle \Lambda_{wp} \rangle$ is the wave packets diffusion coefficient along "+z". Therefore, the propagation of an elementary heat current lasts the time interval $\langle \tau_{wp} \rangle$ and is driven by a virtual temperature gradient $\left\langle \frac{\delta T}{\delta z} \right\rangle$ over the distance $\langle \Lambda_{wp} \rangle$. Of course, in an isothermal medium the number of scatterings along any direction is balanced by that in the opposite one, to get a null diffusion over time. On the contrary, when an external thermodynamic (generalized) force is applied to the system, as for instance a temperature gradient, a heat flux (in the opposite direction) is generated, and therefore, an increase of the number of "*wave-packet ↔ liquid particle*" interactions in the same direction. In this case (Soret effect), events of type (a) of Figure 3 have a larger probability to occur than events of type (b). Analogously, if the symmetry is broken by a particle concentration gradient (Dufour effect), the events having a larger probability of occurrence will be the (b). To quantify such reasoning, let $\frac{dT}{dz} = \frac{T_1 - T_2}{L}$ be the temperature gradient externally applied to a system. As before, we assume here as a first approximation that the application of an external temperature gradient does not change the average total number $\mathcal{N}^{wp}$ of the wave-packets, (small gradient assumption). Let then $\langle \nu_p \rangle$ be the average number per second of "*wave-packet ↔ liquid particle*" collisions at $T_0 \equiv \frac{T_2 + T_1}{2}$ in the isothermal liquid; due to the presence of the virtual temperature gradient, which is equally present in all three directions, for each direction this number is $\langle \nu_p \rangle / 6$. If an external (small) temperature gradient $\frac{dT}{dz}$ is applied to the liquid, the total number of collisions per second $\langle \nu_p \rangle / 6$ is assumed to stay constant because we have supposed that $\mathcal{N}^{wp}$ does not change, but for collisions occurring in the direction of heat propagation, there will be $\delta \langle \nu_p \rangle$ excess of collisions per second and an equal defect $-\delta \langle \nu_p \rangle$ in the opposite direction. If the intensity of the temperature gradient is not too high, we may assume that there will be no non-linear effects and that the initial state of the system at the microscopic level locally continues to be similar to the one at uniform temperature, except for thermal excitations along *z*. Therefore a *liquid particle* at a given place in the liquid experiences the same number of collisions per second as if the temperature was uniform; the only difference with the isothermal case is that there is an imbalance in the number of collisions with wave-packets that originated in the two half-spaces along z, one hotter and the other cooler. In other words, while phonons in liquids are pushed by the local virtual temperature gradient $\left\langle \frac{\delta T}{\delta z} \right\rangle$, the external temperature gradient $\frac{dT}{dz}$ has the duty of orienting them along its own direction. In the linear range, we may assume that $2 \cdot \delta \langle \nu_p \rangle$ is proportional to the temperature gradient $\frac{dT}{dz}$ externally applied to the liquid. Consequently, there is a heat flux $J_q^{ext} = -K_l \frac{dT}{dz}$ superimposed to two equal and opposite heat fluxes $+j_{+z}^{wp} = -K^{wp} \left\langle \frac{\delta T}{\delta z} \right\rangle_+$ and $j_{-z}^{wp} = -K^{wp} \left\langle \frac{\delta T}{\delta z} \right\rangle_-$ ($K_l$ represent

the thermal conductivity of the macroscopic system, $K^{wp}$ is that related to the collective DoF, [17,19]) giving:

$$\frac{2 \cdot \langle \delta v_p \rangle}{2 \cdot \langle v_p \rangle / 6} = \frac{dT/dz}{\langle \delta T / \delta z \rangle} \tag{22}$$

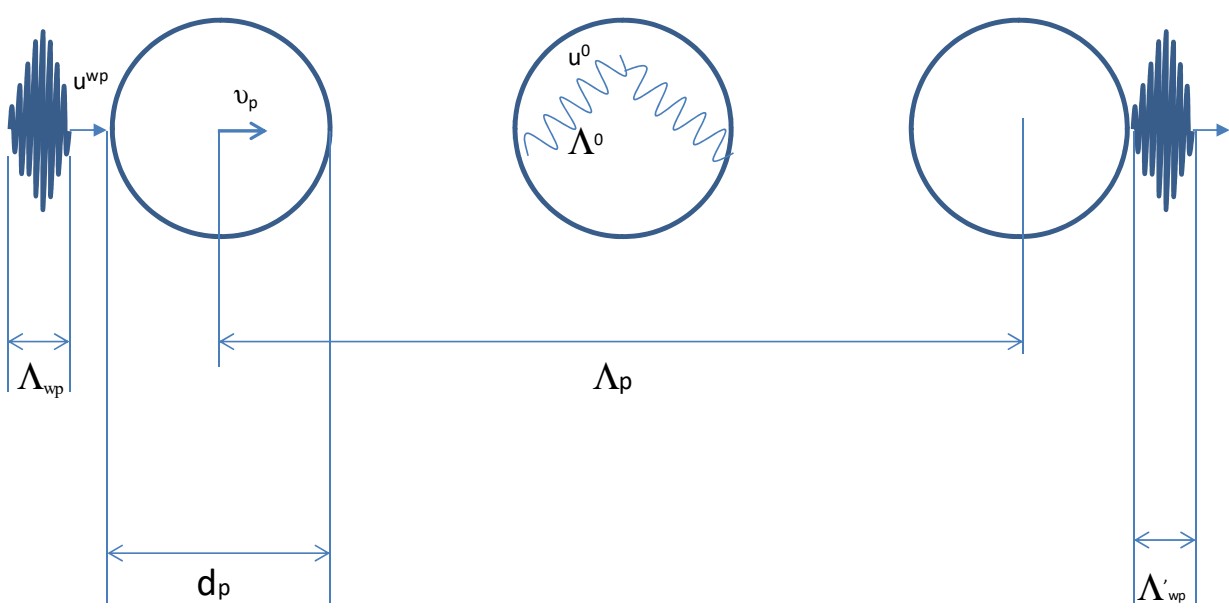

$$\left\langle \Delta^{ph} \right\rangle = \left\langle \Lambda_{wp} \right\rangle + \left\langle d_p \right\rangle + \left\langle \Lambda_p \right\rangle + \left\langle \Lambda'_{wp} \right\rangle$$

$$\left\langle \tau^{ph} \right\rangle = \frac{\left\langle \Lambda_{wp} \right\rangle}{u^{wp}} + \frac{d_p}{u^0} + \frac{\left\langle \Lambda_p \right\rangle}{u^0} + \frac{\left\langle \Lambda'_{wp} \right\rangle}{u^{wp}}$$

**Figure 4.** Close-up of the wave-packet–*liquid particle* interaction shown in Figure 3a. Only the first part of the interaction is represented, i.e., that during which the wave packet transfers momentum and energy to the liquid particle.

Accordingly, an external temperature gradient $\frac{dT}{dz}$ increases by $\frac{dT/dz}{\langle \delta T / \delta z \rangle}$ the phonon flux with respect to that corresponding to the local microscopic heat currents $j^{wp}$ due to spontaneous phonon diffusivity. Equation (22) gives the ratio of the number of "*wave-packet ↔ liquid particle*" collisions per second due to the applied gradient, to that due to the random motions of collective thermal excitations. The clusters therefore will experience $2\delta\langle v_p \rangle$ collisions in excess along the direction of the heat flux and will execute as many jumps per second of average length $\langle \Lambda \rangle$ in excess in the same direction. Consequently, every particle travels the distance $2\langle \Lambda \rangle \delta \langle v_p \rangle$ per second along the direction of heat flux due to the external temperature gradient; this represents the drift velocity $\left\langle v_p^{th} \right\rangle$ of the *liquid particle* along $z$ due to the external temperature gradient:

$$\left\langle v_p^{th} \right\rangle = 2\langle \Lambda \rangle \langle \delta v_p \rangle = \frac{\langle \Lambda \rangle \langle v_p \rangle}{3} \frac{dT/dz}{\langle \delta T / \delta z \rangle} \tag{23}$$

Equation (23) allows defining the thermal diffusion coefficient $D_p^{th}$ that we assume in DML to be the equivalent of $D_T^S$ defined in NET (see Equation (6)) that by definition is the drift velocity of molecules in a unitary temperature gradient:

$$D_p^{th} = \frac{\left\langle v_p^{th} \right\rangle}{dT/dz} = \frac{\langle \Lambda \rangle \langle v_p \rangle / 3}{\langle \delta T / \delta z \rangle} \tag{24}$$

Because $D_p^{th}$ is a signature of the "solute + solvent" mixture, it is correctly given in Equation (24) in terms of the *liquid particle* drift $\langle \Lambda \rangle$ caused by the collisions $\langle v_p \rangle$ with the *lattice particle*. Even more relevant to the purpose of the manuscript is its (reciprocal) dependence upon $\left\langle \frac{\delta T}{\delta z} \right\rangle$. In fact $\left\langle \frac{\delta T}{\delta z} \right\rangle$ is the only term in Equation (24) that intrinsically contains an algebraic sign, determining in such a way the possibility for $D_p^{th}$ to change sign when some particular conditions occur in a liquid mixture. The expression for the Soret coefficient $S_T$ is easily obtained from its definition given by the ratio between the thermodiffusion coefficient and that of ordinary diffusion, for which we exploit the well-known Einstein relation (as specified before, we suppose the mixture sufficiently dilute so that the motion of molecules, either of solute or of solvent, does not interfere each other):

$$S_T = \frac{D_p^{th}}{D_p} = \frac{\langle \Lambda \rangle \langle v_p \rangle}{3 \langle \delta T / \delta z \rangle} \bigg/ \frac{\langle \Lambda \rangle^2 \langle v_p \rangle}{6} = \frac{2}{\langle \Lambda \rangle \cdot \langle \delta T / \delta z \rangle} \tag{25}$$

As for $D_p^{th}$, $S_T$ in Equation (25) contains its algebraic sign hidden in the term $\left\langle \frac{\delta T}{\delta z} \right\rangle$. We will discuss the consequences and implications of Equations (24) and (25) in the Section 5.

A direct evaluation of $S_T$ through Equation (25) is not a simple task, of course, for this reason, dedicated numerical evaluations are in due course and their results will be reported in a separate paper [Peluso, F. et al., in preparation]. However, Equation (25) contains very useful and important information about the meaning of $S_T$ in the DML framework. Recalling that $\langle \Lambda \rangle$ is the *liquid particle* drift caused by the $\langle v_p \rangle$ collisions per second with the *lattice particle*, Equation (25) becomes:

$$S_T = \frac{2}{\langle \Lambda \rangle \cdot \langle \delta T / \delta z \rangle} \approx \frac{2}{\langle \delta T \rangle_z} \tag{26}$$

Here $\langle \delta T \rangle_z$ is the average temperature difference experienced by a solute *liquid particle* over the distance $\langle \Lambda \rangle$ along $z$ covered after a collision with a phonon. This is a direct consequence of the dynamics characterizing the DML and fully in line with the macroscopic meaning of the Soret coefficient in NET.

We now have all the ingredients to calculate the flux of solute (or solvent) molecules $J_m$ due to the collisions with phonons in the presence of an external temperature gradient. If all solute particles have mass $m_p$ and their density is $N_p$, we have:

$$J_m = N_p m_p D_p^{th} \frac{dT}{dz} = N_p m_p \frac{\langle \Lambda \rangle \langle v_p \rangle}{3 \langle \delta T / \delta z \rangle} \frac{dT}{dz} = 2 \cdot \delta \langle v_p \rangle \cdot N_p m_p \langle \Lambda \rangle \tag{27}$$

To recover from the present lack of getting a numerical evaluation of Equation (25), we propose here an analysis of thermodiffusion from another perspective, still in the DML framework, namely that of the variation of $\Pi^{wp}$, or $\Pi^{th}$, along the gradient. Besides being an energy density, $q_T^{wp}$ represents in fact also the pressure $\Pi^{wp}$ exerted by elastic wave-packets on the *liquid particles*. Compiling Equation (13) with Equations (20) and (21) we get:

$$\Pi_{+z}^{wp} \equiv \delta_{+z} q_T^{wp} = \frac{1}{6} \delta q_T^{wp} = \frac{1}{6} \delta \left( \frac{j^{wp}}{u^{wp}} \right) = \delta_{+z} \left( \frac{j^{wp}}{u^{wp}} \right) = \frac{\partial q_T^{wp}}{\partial T} \left\langle \frac{\delta T}{\delta z} \right\rangle_{+z} \langle \Lambda_{wp} \rangle \tag{28}$$

Of course also for $\Pi^{wp}$ the time-averaged pressure generated by the ubiquitous wave-packets streams in a system without external gradients is zero. This is no longer true in the presence of a temperature (or a concentration) gradient working on the system. Assuming as always that the external gradient is small enough that linear deviations suffice to describe the induced variations due to its application, Equation (22) still holds for $\langle \delta v_p \rangle$, i.e., the increase in the number of collisions due to the external gradient. Such an increase in the

number of collisions causes in turn an increase in the energy and momentum exchanged upon the collisions, namely:

$$\delta W^{th} = \langle \delta v_p \rangle \cdot \Delta \varepsilon^{wp} \tag{29}$$

$$\delta F^{th} = \langle \delta v_p \rangle \cdot \Delta p^{wp} \tag{30}$$

The reader has certainly guessed that $\delta W^{th}$ and $\delta F^{th}$ are respectively the total power dissipated by collisions and the force exerted on the collision target, i.e., the *liquid particle* or the *lattice particle*, depending on which situation prevails in the system, (a) or (b) of Figure 3. At the Soret equilibrium, $\delta W^{th}$ is the power needed to sustain the thermal and concentration gradients of the mixture (the Soret equilibrium is characterized by a positive rate of entropy production, the ratio $\delta q/T$ at the cold side being higher than that at the hot side, $\Delta s = \delta q (1/T_c - 1/T_h) > 0$, while $\delta F^{th}$ is the net force responsible for the separation of the two chemical species, for instance, cubes and spheres of Figure 1. Recalling Equations (10), (13), (14), (22), and (30), and adapting Equation (28) for the continuum to the "discrete" situation involving the two distinct *liquid particles* "1" and "2", we get that:

$$\delta F^{th} = \sigma_p \cdot \Delta \Pi^{th} = \frac{\langle v_p \rangle}{6} \frac{dT/dz}{\langle \delta T/\delta z \rangle} \Delta p^{wp} = \sigma_p \cdot \Delta \left( \frac{J_q^{wp}}{u_\phi} \right) = \sigma_p \left[ \left( \frac{J_q^{wp}}{u_\phi} \right)_1 - \left( \frac{J_q^{wp}}{u_\phi} \right)_2 \right] \tag{31}$$

Equation (31) provides a comparison between two expressions for the total force, or the pressure, exerted on the targets following the application of an external temperature gradient. Comparing in fact its third and last members, we understand how the imbalance of the number of collisions influences the dynamical behavior of the mixture. In other words, Equation (31) tells us which of the two species, "1" or "2", is pushed by the collisions to the cold side and why: the chemical species which is pushed towards the cold side by the external thermal gradient is the one for which the difference $\Delta \left( J_q^{wp}/u_\phi \right)$ is positive, and therefore generates a virtual thermal gradient $\left\langle \frac{\delta T}{\delta z} \right\rangle$ parallel to the external one, determining in turn the algebraic sign of $\left\langle \frac{\delta T}{\delta z} \right\rangle$ in Equation (25) or Equation (26). Because the concentration of the solute, i.e., of the less concentrated species, is usually observed in solutions, one refers to positive or negative Soret depending on its concentration variation. Actually, when there is a negative Soret coefficient, according to the DML it happens that $\delta F^{th}$, or $\Delta \Pi^{th}$, is positive for the solvent molecules, which will therefore concentrate on the cold side, while the solute will concentrate on the hot side due to Newton's third law. In general, saying that $\delta F^{th}$, or $\Delta \Pi^{th}$, is positive (negative) means in the DML that the radiation pressure due to the *lattice particle* stream on species "1" is higher (lesser) than on species "2". Because the system is closed, it will obviously happen that only one of the two species will be pushed towards the cold side by the stream of *lattice particles*, while the other chemical species will move to the hot side due to Newton's third law (a bit like cork floats on water while also being attracted by gravity, or a foam floating on the liquid surface will concentrate towards the axis of rotation in a closed rotating system).

In order to get an expression for $S_T$ that can be easily evaluated on an experimental or numerical basis, we write Equation (31) at equilibrium replacing the expression for the thermal flux $J_q$ using the Fourier law; we get then:

$$\delta F^{th} = \sigma_p \cdot \Delta \Pi^{th} = \sigma_p \cdot \left[ \left( \frac{J_q^{wp}}{u_\phi} \right)_1 - \left( \frac{J_q^{wp}}{u_\phi} \right)_2 \right] = \sigma_p \cdot \left[ \left( \frac{KdT/dz}{u_\phi} \right)_2 - \left( \frac{KdT/dz}{u_\phi} \right)_1 \right] \tag{32}$$

where $K$ is the thermal conductivity. Proceeding now in the reasoning, what happens at the equilibrium, i.e., when also the concentration gradient is established in the mixture, is that the power generated by the thermal engine is dissipated against the viscous forces:

$$W^{ph} \equiv W^{th} = W^\eta \tag{33}$$

The velocity of the liquid particle is $\left\langle v_p^{th} \right\rangle$ defined by Equation (23); accordingly:

$$\begin{cases} W^{th} = \delta F^{th} \cdot \left\langle v_p^{th} \right\rangle = \sigma_p \left[ \left( \frac{KdT/dz}{u_\phi} \right)_2 - \left( \frac{KdT/dz}{u_\phi} \right)_1 \right] \cdot \left\langle v_p^{th} \right\rangle \\ W^{\eta} = 6\pi\eta r_p \left( \left\langle v_p^{th} \right\rangle \right)^2 \end{cases} \Rightarrow \quad (34)$$

$$\sigma_p \left[ \left( \frac{KdT/dz}{u_\phi} \right)_2 - \left( \frac{KdT/dz}{u_\phi} \right)_1 \right] = 6\pi\eta r_p \left\langle v_p^{th} \right\rangle = 6\pi\eta r_p D_p^{th} \left. \frac{dT}{dz} \right|_{ext} \quad (35)$$

In Equation (35) we have used Equation (24) and have indicated with the suffix *ext* the external temperature gradient. It is trivial to get:

$$D_p^{th} = \frac{\sigma_p \left[ \left( \frac{KdT/dz}{u_\phi} \right)_2 - \left( \frac{KdT/dz}{u_\phi} \right)_1 \right]}{6\pi\eta r_p \left. \frac{dT}{dz} \right|_{ext}} \quad (36)$$

Introducing the well-known Stokes–Einstein expression for the diffusion coefficient $D_p = \frac{K_B T_{av}}{6\pi\eta r_p}$ eventually we get for $S_T$:

$$S_T = \frac{\sigma_p \left[ \left( \frac{KdT/dz}{u_\phi} \right)_2 - \left( \frac{KdT/dz}{u_\phi} \right)_1 \right]}{K_B T_{av} \left. \frac{dT}{dz} \right|_{ext}} = \frac{\delta F^{th}}{K_B T_{av} \left. \frac{dT}{dz} \right|_{ext}} \cong \frac{\sigma_p \left[ \left( \frac{K}{u_\phi} \right)_2 - \left( \frac{K}{u_\phi} \right)_1 \right]}{K_B T_{av}} \quad (37)$$

Equations (25) and (37) provide for the first time a simple interpretation of the Soret effect at the mesoscopic level, they are without matching parameters and can be verified by means of experiments. Equation (25) is based on elementary principles applied at the mesoscopic level, while Equation (37) is based on macroscopic quantities. Both equations and their implications, as well as their capabilities in the comparison with experimental data on the Soret coefficient and on its sign, will be extensively analyzed in the Section 5.

## 4. An "Unexpected" Mechano-Thermal Effect in Pure Liquids Explained by Means of DML

The other cross-effect we will frame in the DML is that discovered by the group of Noirez a few years ago [6–11]. The effect results from the coupling between an acceleration field and a thermal field in a confined liquid layer submitted to shear strain. The system consists of a pure liquid layer confined in between two coaxial disc plates, one is fixed while the other one is mobile following a Heaviside function or an oscillatory motion, see Figure 5 for a schematic. The system is observed by far by means of a thermocamera, which allows for measurement of the thermal field that is being established inside the liquid, as well its time evolution. When the Heaviside motion law is applied to the mobile plate, typically the lower one, it undergoes a sudden acceleration in one direction, performs the expected movement, and then stops for a few seconds. Then it goes back to the starting point with an acceleration lower than that of the outward path. In the second type of experiment, the rotating plate oscillates according to a periodic law. The thermal behavior of the liquid in the two cases is not exactly the same, although the dependence of the intensity of the mechano-thermal phenomenon, in particular of the temperature difference that originates with respect to the acceleration field, remains almost the same. From the classical point of view, even a hardly detectable (faint) heating of the liquid layer facing the moving plate is not expected. Indeed, at these low frequencies and strain amplitudes, the energy stored cannot generate viscous heating following the classical empirical evaluation of the Nahme number (also known as the Brinkman number). Much faster velocities are needed for viscous liquids, closer to sound velocities. Experiments performed by Noirez and co-workers revealed instead that unexpectedly liquids confined to sub-millimeter scale show a thermal response consisting of the cooling of the layer facing the moving plate and

a corresponding heating of that facing the fixed plate. The aim of this section is to provide a theoretical interpretation of such counter-intuitive response in the framework of DML.

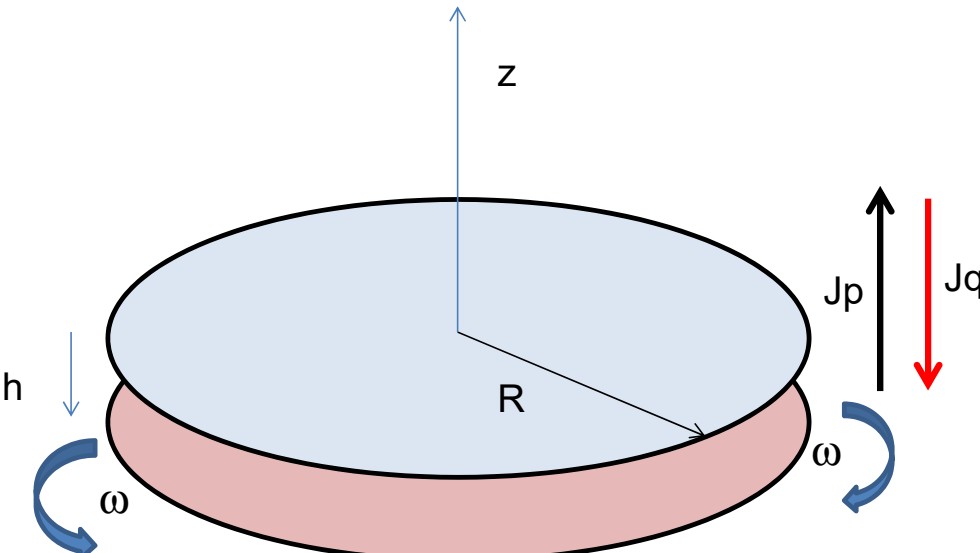

**Figure 5.** Schematic (drawing not to scale) of the experimental device adopted in [11]. The liquid, glycerol in this case, lies in between the two plates, one of which may rotate, the other is fixed.

Generally, the non-uniform temperature shear bands observed along the strain direction indicate the formation of a faint temperature gradient in phase with the deformation. The synchronization of the thermal field with the mechanical strain is a signature of a mechano-thermal effect of elastic origin indicating that viscous liquids might exhibit thermoelastic behavior, as pointed out also by the authors of the experiments [108,109].

Noirez and co-workers pointed out that this response to low-frequency mechanical stress is typical of amorphous solids, exhibiting a storage shear modulus $G'$ much higher than the loss viscous modulus $G''$. This is a key point in both of the experiments and the theoretical interpretation of the DML framework that we are going to show in what follows. In fact, being the geometry of these experiments of shear strain type, it is implicitly demonstrated by the experimental results that liquids possess shear elasticity as solids, contrary to what is generally supposed due to the absence of rigidity, but in line with the duality of liquids as supposed in the DML. By increasing the frequency of the mechanical stress and/or the scale on which such stress is applied, the effect disappears, and the liquid returns to its classical full-viscous behavior ($G' = 0$). Nonaffine framework has provided the physical interpretation of such behavior, consisting of the cut-off of low-frequency transverse modes, that in turn allows the high-frequency modes to maximize their effect at the mesoscopic level [16]. In this framework, the authors have assumed that transverse modes are carried by phonons. Accordingly the DML may provide the missing ingredient to nonaffine treatment by means of the phonon $\leftrightarrow$ liquid particle interactions, where phonons assume the role of transforming the momentum they receive from the liquid particles in a thermal field. Therefore, $f^{th}$ is the best candidate for the nonaffine force introduced in the equation of motion of the microscopic building block [16]. In fact, in the nonaffine framework, there is a competition between the rigid-like behavior of coordinated clusters and the relaxations via nonaffine motions due to unbalanced forces in the disordered molecular environment.

Let's then proceed with the DML interpretation. The key is provided indeed by the elementary *lattice particle $\leftrightarrow$ liquid particle* interaction, described in Figure 3b. The acceleration of the moving plate generates a velocity gradient through the liquid, which in turn gives rise to a momentum flux, directed towards the fixed plate, following the relation $J_p = -\eta_l \frac{dv}{dz}$, with $\eta_l$ the liquid viscosity. According to the DML, and also following

Rayleigh's reasoning, the momentum flux is dimensionally an energy density and generates a temperature gradient. The *liquid particles*, pushed by the momentum flux, transfer their excess impulse to the phonons of the liquid, following the (b) event of Figure 3; the phonons emerge from each single impact with an increased energy, being frequency-shifted. They are then pushed and collected towards the fixed plate, giving rise to a temperature gradient directed towards the fixed plate. The rationale is the same adopted to justify Equation (22), where however in this case the imbalance of collisions $\langle \delta v_{wp} \rangle$ affects the wave packets instead of the *liquid particles*, and is due to the ratio between velocity gradients instead of temperature:

$$\frac{2\langle \delta v_{wp} \rangle}{2\langle \delta v_{wp} \rangle / 6} = \frac{dv/dz}{\langle \delta v / \delta z \rangle} \tag{38}$$

The supposed linear dependence this time is between the external velocity gradient $\frac{dv}{dz}$ and that virtual, normally present in mechanical equilibrium for a *liquid particle* between two successive collisions with the *lattice particles*, $\left\langle \frac{\delta v}{\delta z} \right\rangle$. Each wave packet executes $2\langle \delta v_{wp} \rangle$ jumps per second in the direction of the momentum flux, each long $\langle \Lambda^{wp} \rangle$; they accumulate towards the fixed plate and decrease near the moving one, generating the temperature field observed. The product

$$v_{\nabla v}^{wp} = 2\langle \Lambda^{wp} \rangle \langle \delta v_{wp} \rangle \tag{39}$$

defines the drift velocity of the thermal front due to the velocity gradient. The phenomenon is therefore very rapid because it is determined by a succession of (almost) elastic collisions, which propagate very quickly. Obviously, the higher thermal content of the liquid in proximity to the fixed plate compensates for the lower thermal content near the rotating plate, keeping the thermal balance unchanged (neglecting the losses toward the environment). It is important to highlight that the *lattice particle* $\leftrightarrow$ *liquid particle* collision also explains why a temperature gradient (i.e., a generalized flux-force crossed effect) is generated inside the liquid instead of a simple heating due to the energy stored as a consequence of the mechanical stress ($G' >> G''$), as one would have expected on the basis of a classical interpretation of the elastic modulus. At the same time, the phenomenon can be observed only on the small length scale as that adopted in the experiments, the faintness of the thermal gradient would be in fact destroyed by the thermal unrest of the molecules in larger liquid volumes.

When the plate stops, the system relaxes and reaches the temperature it had at the beginning. During the return of the rotating plate to the starting position, the liquid is in exactly the same condition it had at the beginning of the experiment, and the temperature gradient forms again. This interpretation is in line with all the basic hypotheses made by the authors, including the one on the number of Grüneisen. In this particular case, the expansion/compression effect of the material medium would be a consequence of the density of the phononic contribution.

We will now try and get an expression for the (rate of) temperature variation due to the occurrence of events of type (b) of Figure 3 which are at the base of the observed phenomenon. In fact, what happens inside the liquid is that the mechanical energy injected into the liquid by the rotating plate is (partially) commuted in thermal energy carried by wave-packets. This energy, as we have seen, accumulates close to the fixed plate. To do this, we begin by considering that the momentum flux $J_p$ through the liquid due to the plate rotation represents also the pressure $\Delta\Pi_l$ exerted by the liquid molecules on the wave-packets, and is given by [110]:

$$J_p = \left(\rho v^2\right)_l = \Delta\Pi_l \tag{40}$$

where $\rho_l$ and $v_l$ are the density and shear speed of liquid, respectively. Neglecting dissipation effects due to the liquid viscosity, and remembering Equation (13), the momentum flux

is converted into the pressure that pushes the phonon's stream and gives rise to a heat flux along the opposite direction:

$$J_p = \Delta\Pi_l \equiv -\Delta\Pi_{th} = -\Delta\left(\frac{J_q^{wp}}{v^{wp}}\right) \tag{41}$$

where $v^{wp}$ is the wave-packet average speed. As is well known, the momentum flux corresponds even to the energy density $\Delta W^{th}$ injected into the system by the mechanical stress. The duality of the DML provides us a key to (partially) convert at a mesoscopic level such mechanical energy in thermal energy, i.e., that carried by phonons that are excited at higher frequencies upon interactions of type (b) of Figure 3 with the *liquid particles.* Neglecting variations with respect to the temperature, we may use the average macroscopic value given by the ratio $v^{wp} = h/\Delta t$, where $h$ is the height of the liquid sample (parallel to the rotation axis), and $\Delta t$ the time necessary to establish the observed temperature difference across the liquid. Such value will be a multiple of the ratio $\langle\Lambda\rangle/\langle\tau\rangle$. Getting now the temperature difference $\Delta T$ (or the temperature gradient $\Delta T/h$) is trivial, being enough to divide Equation (41) by the specific heat (per unit of volume) of the substance, i.e.,:

$$\Delta T = \frac{1}{(\rho C)_l}\Delta\left(\frac{W^{th}}{v^{wp}}\right) = \frac{1}{(\rho C)_l}\Delta\left(\frac{J_q^{wp}}{v^{wp}}\right) \approx \frac{v_l^2}{C_l} \tag{42}$$

Using the data contained in [11], we give now a numerical estimation of $\Delta T$ from Equation (42). First, we use the value of the momentum flux averaged over the disc surface, $\langle J_p \rangle = \frac{1}{A}\int_0^R (\rho v)_l^2 dA = \frac{\rho_l \omega^2 R^2}{2}$, where $\omega$ is the plate angular velocity and $R = 20$ mm its radius. The liquid used in the experiment was glycerol, for which $\rho_l = 1.26$ g/cm$^3$ and $c_p = 2.39$ J/g·K. Using a value of 100 μm for $h$ and 10 Hz for $\omega$, one gets $\Delta T \approx 0.06$ K of the correct order of magnitude as measured and reported in [11].

It is relevant to point out that the above interpretation is made possible because the DML foresees the presence of wave-packets that, interacting with the *liquid particles*, behave in this case not only as carriers of the thermal energy but also as converters of mechanical energy into thermal energy.

## 5. Discussion

In this paper, we have used the DML to explain two cross effects in liquid mixtures and in a pure liquid out of equilibrium, namely the thermodiffusion and a mechano-thermal effect recently detected in isothermal liquids in shear strain geometry. Both have been interpreted thanks to the duality of liquids, supposed in the DML as constituted by metastable solid-like icebergs fluctuating in the liquid amorphous phase [17]. Elastic perturbations, hence the heat, are carried by elastic wave-packets that interact with the *liquid particles* exchanging with them energy and momentum. In particular, the exchange of momentum allows to provide an interpretation of the selective migration of solute and solvent molecules, which are pushed along the thermal current, i.e., the current of wave-packets, following the dynamics shown in Figure 3a. More in detail, the separation of the components of a liquid mixture upon the application of a thermal gradient, is due to the capability of the forces acting at the mesoscopic level in a liquid to convert the heat flux into a momentum flux. An external temperature gradient has the effect of orienting the local gradients along the same direction. When instead $S_T$ or $D^{th}$ are negative means that the local virtual gradients are oriented along the direction opposite to that of the external gradient (see Equations (25) and (37)). This circumstance produces a thermodiffusive drift along the direction opposite to that of the external gradient, resulting in a negative value of $S_T$. In other words, the thermodiffusive drift of the solvent prevails on that of the solute, consequently we will observe the solute migrating to the hot side and the solvent to the

cold one because the force due to the radiation pressure exerted by wave-packets on the solvent particles is higher than that produced on the solute particles.

Through the equations describing the DML and the *liquid particle ↔ lattice particle* interactions, we have provided for the first time an expression (actually two) for the Soret coefficient $S_T$ without matching parameters, falsifiable (in the Popperian meaning) because verifiable through experiments and manifesting even the sign dependence typical of the Soret coefficient. The order of magnitude of $S_T$ as well the sign inversion will be compared in this section with experimental data available in the literature. The time-reversal mechanism shown in Figure 3b is instead responsible, in our vision, for the Dufour effect and of the mechano-thermal effect described in Section 4.

In this section, we will discuss the equations and deepen the concepts introduced in the previous two paragraphs. Let's then start with the expressions for $S_T$. Equation (26) allows us to speculate on a very intriguing aspect related to the meaning and definition of $S_T$ in NET. As widely discussed in [17,19], in the phonon-*liquid particle* collisions of Figure 3a, part of the phonon energy is transformed into kinetic and potential energies of the *liquid particle*. The interaction definitively has an activation threshold for potential energy and, depending on how much energy is absorbed by internal DoF, the rest of the transferred energy will increase the kinetic energy of the *liquid particle*. When an external temperature gradient is applied to a system, the first effect that occurs is the establishment of the temperature gradient across the system, according to the Cattaneo–Fourier equation [19]. Therefore, once the internal DoF have reached the statistical equilibrium in terms of the distribution of their degree of excitation depending on the local temperature, the phenomenon of matter diffusion takes place due to the temperature gradient and at the expense of the kinetic energy reservoir, Equation (21). Inelastic effects dominate the dynamics during the transient phase, leaving the control to the kinetic reservoir at the steady-state, where only the elastic effects of the elementary interactions matter. Accepting the above reasoning $S_T$ may be seen as the activation energy for the thermodiffusion process:

$$C(T) = C_0 \exp\left(-\frac{R/S_T}{RT}\right) = C_0 \exp\left(-\frac{1}{TS_T}\right) \approx C_0 \exp\left(-\frac{\langle \delta T \rangle_z}{T}\right) \qquad (43)$$

where $C$ is the solute concentration: if the energy $RT$ available at the temperature $T$ exceeds the activation energy $R/S_T$, or alternatively, the average temperature of the system exceeds the temperature difference over $\langle \Lambda \rangle$ following the elementary interaction (see Equations (25) and (26)), the phenomenon takes place. It should be interesting to theoretically speculate on whether, and how, such activation energy is related to the presence in liquids of *gapped momentum states* [111,112]. We have already discussed in [17] how the "*lattice particle ↔ liquid particle*" interaction can be assumed as responsible for such phenomenon in liquids exhibiting *k-gap*.

Of the two expressions obtained for $S_T$, Equation (25) is based on elementary principles applied at the mesoscopic level, while Equation (37) is based on macroscopic quantities. A question that the reader has certainly wondered is what one expects from their comparison. Let's then consider the third member of Equation (37); the numerator is the force exerted by the thermal field to displace the solute or solvent, molecules to generate and sustain the concentration gradient that gives origin to the Soret effect. At equilibrium the (average) energy available in the system is $K_B T_{av}$. The ratio between such two quantities is just the mean displacement of the solute (solvent) *particle*, $\langle \Lambda \rangle$, due to the collisions with the phonons, $S_T = \frac{\delta F^{th}}{K_B T_{av} \frac{dT}{dz}\big|_{ext}} \approx \frac{1}{\langle \Lambda \rangle \cdot dT/dz} \approx \frac{1}{\langle \delta T \rangle_z}$. Consequently, Equations (25) and (37) formally represent two similar expressions for the Soret coefficient (apart from a numerical factor due to statistical averages).

Equation (37) allows us to open a very interesting discussion, based on several points. The first and more immediate is that the last member is obtained by simplifying the temperature gradient in the two parentheses in the numerator with that in the denominator. Equation (37) has a huge relevance because it allows it to evaluate $S_T$ more easily than

Equation (25). Nevertheless, the difficulty in this case arises from the fact that the physical quantities involved are not, in principle, those of the bulk materials, but rather those in the effective configuration in which they are in the mixture. What helps is that in the DML, solvent, and solute *particles* are not seen as isolated molecules but rather as islands, those we have dubbed *icebergs* or *liquid particles*. Incidentally, the proportionality of Equation (37) to the cross-section of the solute particle copes with the law $S_T \propto M_p^{2/3}$ to the solute mass early evidenced by Emery and Drickamer [46] and successively by other authors [40,52] for solutes of large molecular weight. For such large particles, in fact, the surrounding layer of solvent molecules is relatively less relevant than for small solute particles.

In Table 1 some examples of Soret coefficients evaluated from Equation (37) are provided for both positive and negative $S_T$. Table 2 collects data for the rations $K/u$ elaborated from thermodiffusion in liquid mixtures performed with a Klusius–Dickel device. Original data report only which of the two species involved in each single experiment concentrate on the hot or cold side, however, this is enough to verify how the direction of the thermodiffusion drift is correctly predicted from Equation (37).

**Table 1.** Experimental values for Soret coefficient (last column, units $10^{-3} \text{K}^{-1}$) for mixtures of polyvinylpyrrolidone K90 of 360,000 amu in various solvents as listed in first column, measured and reported in [113]. Thermodiffusion of this macromolecule was experimentally studied in several solvents; in particular, an inversion of the sign of the Soret coefficient was detected in buthanol and propanol. The columns 2 and 3 report the values of the ratios $K/u$ for the several substances (units $10^{-3} \text{ Jm}^{-2}\text{K}^{-1}$). Their differences are reported in fourth column. It is interesting to note the sign inversion for the same two solvents as experimentally detected. Data for $K's$ and $u's$ are deduced from [114].

| Mixture | $\left(\frac{K}{u}\right)_l$ | $\left(\frac{K}{u}\right)_p$ | $\left(\frac{K}{u}\right)_l - \left(\frac{K}{u}\right)_p$ | $S_T$ |
|---|---|---|---|---|
| K-90 in water | 0.4031 | 0.137 | 0.266 | 19.82 |
| K-90 in methanol | 0.1830 | 0.137 | 0.046 | 0.38 |
| K-90 in ethanol | 0.1440 | 0.137 | 0.007 | 2.31 |
| K-90 in buthanol | 0.1205 | 0.137 | −0.016 | −5.78 |
| K-90 in propanol | 0.1197 | 0.137 | −0.017 | −6.01 |

**Table 2.** Values of the ratios $K/u$ elaborated from experimental data of thermodiffusion in liquid mixtures obtained with the Klusius-Deckel device [115]. The columns 2 and 3 report the values of the ratios $K/u$ for the substances (units $10^{-3} \text{ Jm}^{-2}\text{K}^{-1}$). Their differences are reported in fourth column. It is interesting to note the sign inversion for the same two solvents as experimentally detected. Data for $K's$ and $u's$ are deduced from [114]. Data from the two Tables, in particular for the ratios $K/u$, must however be taken *cum grano salis* and considered with caution, a thorough evaluation of $S_T$ in the DML should be elaborated from Equation (25).

| Mixture | $\left(\frac{K}{u}\right)_1$ | $\left(\frac{K}{u}\right)_2$ | $\left(\frac{K}{u}\right)_1 - \left(\frac{K}{u}\right)_2$ | Component Drifting to Cold Plate |
|---|---|---|---|---|
| Hexane-Carbontetrachloride | 0.1130 | 0.1120 | 0.0010 | 2 |
| Hexane-Cyclohexane | 0.1130 | 0.0978 | 0.0152 | 2 |
| Toluene-Cyclohexane | 0.1100 | 0.0978 | 0.0122 | 2 |
| Toluene-Benzene | 0.1100 | 0.1028 | 0.0072 | 2 |
| Benzene-Chlorobenzene | 0.1028 | 0.1021 | 0.0007 | 2 |
| Benzene-Nitrobenzene | 0.1028 | 0.0990 | 0.0038 | 2 |
| Bromobenzene-Carbontetrachloride | 0.0973 | 0.1120 | −0.0157 | 1 |
| m-Xylene-o-Xylene | 0.0978 | 0.0974 | 0.0004 | 2 |

Figure 6 shows another way to conceptually validate the expression of $S_T$ given by Equation (37). Figure 6a collects the data obtained by Bierlein, Finch, and Bowers for several mixtures of benzene and n-heptane [116]. The interesting point is that $S_T$ for such liquid mixtures increases with the average temperature for all the relative concentration values of the two components. Even more interesting is the fact that the three plots obtained as a function of temperature cross the "zero" value for $S_T$ at about the same temperature, around 60 °C. More precisely, $S_T$ is negative for temperature below such threshold, goes through zero, and becomes positive for average temperature above the threshold. In Figure 6b we have plotted the values of the ratio $K/u$ for the two components; it is straightforward to note that such a ratio assumes the same value around the temperature of 60 °C. Because the sign of $S_T$ in Equation (37) depends just upon the difference of the ratios $K/u$ for the two components, this observation confirms that Equation (37) is a good candidate to foresee even the sign inversion of $S_T$. As specified above and also in Section 3, Equation (25), that is in principle the correct expression for $S_T$ in the DML, contains the sign dependence hidden into the term $\langle \delta T / \delta z \rangle$, more precisely in the orientation of $\langle \delta T / \delta z \rangle$ with respect to the external temperature gradient. Supporting the above reasoning is the dependence of $D_p^{th}$ in Equation (25) upon $\langle \delta T / \delta z \rangle$ instead of the external gradient. This is not surprising because the thermodiffusion is a signature of the mixture, therefore it shall be dependent only upon the mixture's parameters. Moreover, the reader must not be fooled by the inverse proportionality with $\langle \delta T / \delta z \rangle$: it tells us that as large the local virtual gradient $\langle \delta T / \delta z \rangle$, as difficult will be to observe the thermodiffusion, as one would have expected. Of course, $\langle \delta T / \delta z \rangle$ as well $\langle \Lambda \rangle$ or $\langle v_p \rangle$ will change according to the relative concentration of the components, or to the temperature of the system.

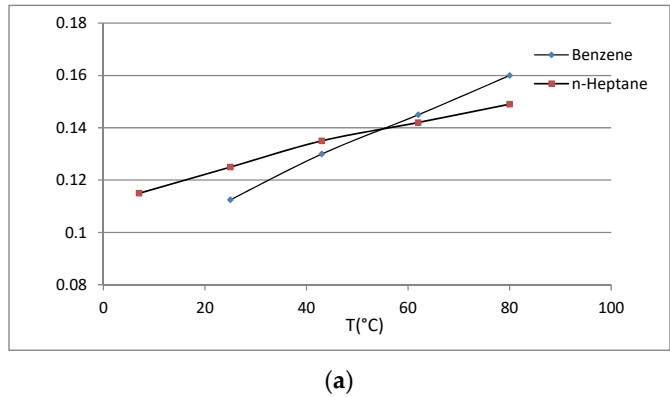

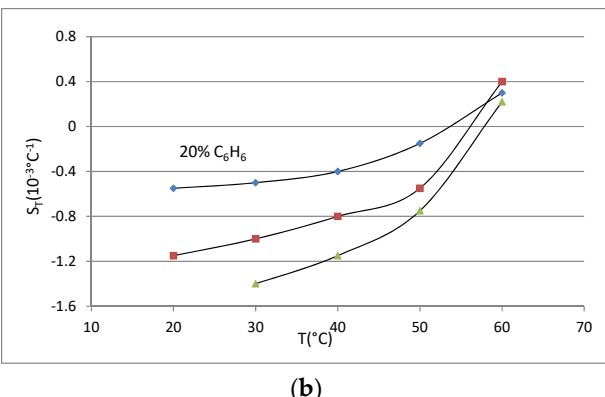

(**a**)          (**b**)

**Figure 6.** (**a**) Values of Soret coefficient deduced from [116] and their best fit lines. (**b**) Values of the ratios $K/u$ and their best fit lines for the same couple of liquids as in (**a**). The two plots cross at about the same temperature as where $S_T$ crosses the "zero" in (**a**).

A more extensive collection of experimental data on thermodiffusion is in due course and their comparison with the theoretical predictions of Equations (25) and (37) will be shown elsewhere. Because we do not have access to a lab, the only possibility to provide further quantitative evaluations of Equation (37) is through experimental data available in the literature for which enough data should even be available for $K$'s and $u_\phi$'s. As it concerns the validation of Equation (25), dedicated numerical analysis will be the core of another work.

The comparison between Equations (25) and (37) opens to a more wide reasoning concerning the displacement of the energy and mass barycenters of a mixture as a consequence of the thermodiffusion. Making reference here to Figure 7, the drawing on the left shows a mixture, "0", constituted by two chemical species, "1" and "2", in a container in thermal equilibrium at $T_0$ with the environment; only the exchange of thermal energy is allowed with the environment. For such a system the mass, $S_0$, $S_1$, $S_2$, and energy, $E$, barycenters are all coincident and positioned in the cell middle. Figure 7b on the right shows the

same system once the Soret equilibrium is reached. In this situation, we have a heat flux $J_q = -K\frac{dT}{dz}$ crossing the system, with $T_w > T_0 > T_c$, so that the four barycenters are no longer coincident. In the drawing we have supposed "1" to be the solute, with a molecular mass $\rho_1$ and ratio $(K/u)_1$ higher than those of the solvent "2", $\rho_2$ and $(K/u)_2$; this explains the positioning of the three mass barycenters. As it concerns the energy barycenter $E$, it is shifted towards the warmer side due to the higher thermal content of the system. Several considerations are now in order. (i) The force acting on the mixture's components pushes the solute particles toward the cold side; consequently, the solvent particles are pushed toward the warm side. The fact that $S_2$ is shifted toward the opposite side with respect to $S_1$, is a direct consequence of the III Principle of dynamics, being the system closed, so that a net volume flux is not allowed, $J_V = 0$. This is a relevant point because some authors in the past have argued that the phenomenon of thermodiffusion was characterized by a violation of the III Principle [47]. (ii) The energy introduced into the system at the expense of the external thermal field ensures the stability of the mass and energy distributions obtained as effects of the thermodiffusion and gives also rise to a positive rate of entropy production. However, to keep such distributions stable with time, it is necessary that part of this energy is converted into momentum, which pushes the solute and solvent molecules towards their respective barycenters (see Equations (29) and (30) and related comments). Let's then calculate this momentum rate, for which we will see that the duality of liquids of the DML is a necessary prerequisite, being the phonons the carriers of such momentum flux. If $\Delta t$ is the time interval to reach the Soret equilibrium, the net mass current $J_M$ will be given by (Figure 7b):

$$J_M = (\rho_1 - \rho_2) \cdot \frac{\Delta z}{\Delta t} \cdot A = (\rho_2 - \rho_1) \cdot D_1^{th} \frac{dT}{dz} \cdot A \tag{44}$$

where $A$ is the cell cross area and $D_1^{th}$ the coefficient of thermal diffusion of species "1". The mechanical momentum flux, $\frac{d(mv)}{dt} \equiv v\frac{dm}{dt}$ at equilibrium, is then given by:

$$v\frac{dm}{dt} = \frac{\Delta z}{\Delta t} \cdot J_M = \frac{\Delta z}{\Delta t} \cdot (\rho_1 - \rho_2)\frac{\Delta z}{\Delta t}A = (\rho_1 - \rho_2) \cdot \left(D_1^{th}\frac{dT}{dz}\right)^2 A \tag{45}$$

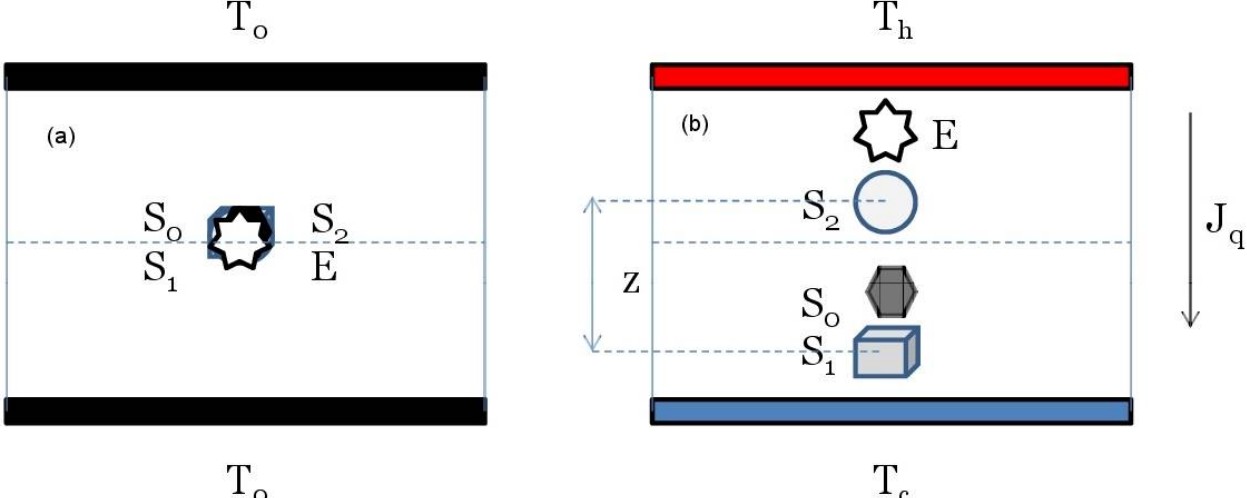

**Figure 7.** (**a**) Closed system constituted by a mixture of two substances at thermal equilibrium with the environment; only thermal energy can be exchanged with the environment. (**b**) How the mass $S_0$, $S_1$, $S_2$ and energy $E$ barycenters of the same system as in (**a**) appear at the Soret equilibrium, by supposing that the species "1" has and higher density than species "2".

Equation (45) gives the net rate of transfer of momentum from the phonons to the solute particles in the course of the processes of type (a) shown in Figure 3.

It is instructive and interesting to perform the same calculation but in a situation in which the system is closed and thermally isolated, and the starting situation is the one in which the two components are already separated, for instance with a solute concentrated solution on the bottom of the cell and the pure solvent on the top. We all know that the final distribution reached by the system will be that of a homogeneous mixture; we could thus think of a time reversal experiment with respect to thermal diffusion, known in NET as the ordinary diffusion giving rise to the Dufour effect. Of course, the constraint $J_V = 0$ is valid also in this case. Equation (44) now becomes:

$$J_M = (\rho_0 - \rho_1) \cdot \frac{\Delta z}{\Delta t} A = (\rho_0 - \rho_1) \cdot D_1 \frac{dn}{dz} A \tag{46}$$

$\rho_0$ is the density of the solution, $D_1$ the solute diffusion coefficient and $\frac{dn}{dz}$ the time averaged value of the solute concentration gradient. The momentum transfer accordingly is:

$$v \frac{dm}{dt} = \frac{\Delta z}{\Delta t} \cdot J_M = \frac{\Delta z}{\Delta t} \cdot (\rho_1 - \rho_2) D_1 \frac{dn}{dz} A = (\rho_0 - \rho_1) \left( \frac{\Delta z}{\Delta t} \right)^2 A \tag{47}$$

This time Equation (47) represents the momentum transfer from the *liquid particles* to the phonons, following the events represented in Figure 3b. The reader may now wonder how it is possible that the momentum of an isolated system has changed. The duality of the DML makes the answer straightforward; in fact, the phonon population was initially uniformly distributed in the system, which was indeed in thermal equilibrium, but not the population of solute particles, for which a concentration gradient is present at the beginning of the experiment. The mass flow due to the concentration gradient produced in turn a phonon concentration gradient opposite to the solute concentration gradient, due to an excess of collisions with solute particles in the direction of the mass flow, i.e., following just the type (b) processes of Figure 3. This quantity is used by the *liquid particles* to shift their barycenter mass. Of course, and as is well known, the Dufour effect produces a (faint) thermal gradient, that we identify as that of the phonon concentration gradient that is established in the system as a consequence of the *liquid particle* current. Once again, the two events shown in Figure 3 represent the elementary interactions occurring in liquids, either in equilibrium or out of equilibrium, their time-reversibility being the best candidate to justify the Onsager reciprocity law at the mesoscopic level [17,19].

The mechano-thermal effect analyzed in Section 4 even deserves some comments. Equation (42), with all the limitations and simplifications adopted, shows that the temperature difference is proportional to the square of the shear speed, making the $\Delta T$ orientation independent upon the shear versus, as expected. The energy and momentum input depend only upon the momentum of the moving disc, and the $\Delta T$ observed is independent of the amount of substance or of the gap thickness of the layer ($C_l$ is the specific heat per unit of volume). In the comparison of our results with the experimental observation, we should on one side keep in mind either the simplifications adopted in obtaining Equation (42), and the experimental difficulties in getting a stable and well-thermally isolated experimental setup. Nevertheless, on the other side, we could speculate on those aspects that may appear contradictory. One of these is that in some cases an apparent inversion of the thermal gradient was observed upon inversion of the shear motion. This effect could have the same origin as that observed in those cases where more than a single gradient was observed in the liquid, a sort of alternation between hot and cold layers. A possible explanation, although on a purely speculative basis, could be that the stationary plate reacts to the inversion of the shear direction with a sort of "mirror" effect, pushing back the thermal excitations. The liquid near the stationary plate perceives the shear inversion as a sort of mirror effect of the stationary plate. Very interestingly, upon the revision process, one of the Reviewers called my attention to the non-Fourier processes recently identified in various thermal mechanisms. The review by G. Chen [117] may help us to understand the origin of such effects. Generally, the momentum of normal phonons is conserved in scattering

processes; in systems where normal phonon scattering dominates, phonons under temperature gradient can acquire a non-vanishing drift velocity similar to fluid flow driven by a pressure gradient: this is named the phonon hydrodynamic regime, which originates from the Landau treatment of superfluid Helium [103], and has been already dealt with in detail in a previous paper on the DML [17]. The Casimir–Knudsen is one among the several heat conduction phenomena associated with the phonon hydrodynamic regime. A necessary (but not sufficient) condition for the Casimir–Knudsen regime to take place is that the phonon mean free path $\langle \Lambda_0 \rangle$ is comparable to, or larger than, the characteristic length of the system, say $\langle l \rangle$. In the system under study, the phonons propagating in the pure liquid interact with the material constituting the plates. Hence the problem is restricted to considering such interface, of thickness $\langle l \rangle$, and the related highly uncertain boundary conditions. These uncertainties arise because of the atomic-scale defects present at the interface affected by physical roughness, in turn responsible for how phonons are scattered because the phonon wavelength amounts to several nanometers, as estimated also in DML ([17], Table 1). To bypass the problem a "specularity" parameter is usually introduced, accounting for the fraction of "specularly" reflected phonons, which should depend on the phonon wavelength and surface roughness [118]. The speculative explanation proposed above could then be a consequence of the shear strain and may represent coherence bands of phonons originated just at the interface between the plate and the liquid. The incoming phonons experience reflection and transmission, being responsible for the temperature drop across the interface. This phenomenon, quantified in Section 4, could explain either the inversion of the temperature drop upon inversion of the plate rotation or the formation of multiple temperature layers.

One of the characteristics of the Casimir–Knudsen regime is that the thermal conductivity is linearly proportional to the specific heat, as is the case just for the thermal conductivity $K_l^{wp}$ in the DML (see Equation (12) in reference [19]). During the thermal transient, the heat pulse generates heat wave propagation (second sound) similar to a mechanical pulse generating a sound wave in a solid. Thus, as extensively described in [19], phonon heat propagation can be described using a Cattaneo-type (telegraph) equation [117]. In addition to thermal transport, it has also been predicted that signatures of phonon hydrodynamic transport can be observed from neutron and Brillouin scattering experiments [17,119]. In conclusion, the DML allows a direct comparison with many other interesting aspects characterizing either the solid-like behavior of liquids or the presence of phonons as carriers of elastic (and thermal) excitations.

Returning to Equation (28) or Equation (31), it is interesting to note that $\Delta \Pi^{th}$ represents the radiation pressure exerted by the current of wave-packets on the *liquid particles* following the interaction. This pressure may also be regarded as a sort of osmotic pressure determining the displacement of *liquid particles* following the collisions with the wave-packets. Indeed, to a pressure difference, $\Delta \Pi^{th}$ the pure liquid responds with the self-diffusion of the *liquid particles* [17,19]. The tunnel effect could be regarded as a semi-permeable membrane, that works allowing only the passage of *liquid particles* and preventing that of *lattice particles*. A similar argument was raised by Ward and Wilks [120] in dealing with the mechano-thermal effect observed in HeII.

Finally, some considerations could be dedicated to a comparison between the capabilities of DML applied to cross effects in NET and the previous theoretical models for thermodiffusion. As anticipated in Section 2, we do not want here to enter into the debates of the models or make a historical excursus, that is however the topic of other interesting papers [12,13], but only propose a critical comparison of the DML with some of the most relevant alternative models. The first comment is dedicated to the interesting paper by Najafi and Golestanian [47], in which they suppose the thermodiffusion to be due to the Soret-Casimir effect. They in fact calculate such force in several configurations, however this method is intrinsically unable to foresee negative values for $S_T$. Besides, they have also argued that in some cases the Soret effect violates the III Principle of dynamics, a circumstance that has been definitively excluded in the present manuscript because of

the liquid's duality. Duhr and Braun [52] have modeled the thermodiffusion due to the gradients of thermodynamic potentials inside mixtures. They have applied the model to macromolecular solutions, in particular DNA, finding good correspondence also with the mass and size of the solute, however only in those cases with positive $S_T$.

Two very interesting historical reviews are represented by the works of Eslamian and Saghir [12] and Rahman and Saghir [13], from which a large amount of models of thermodiffusion jumps to the eye, very different from one another and often from experimental data. These reviews are united by similar conclusions on the several models they deal with. In particular, it is affirmed that many theoretical models, initially accepted by the scientific community, were found wrong after more careful analysis [49,121]; many others correctly reproduce the thermodiffusion only in gases, while "in liquids, reliable and satisfactory predictive theories are still lacking"; furthermore "the largest part of models necessitate of external matching parameters to be defined, making the models dependent on the experiments" [12]. A relevant point is the fact that quite all the models either do not predict negative $S_T$ or fail in its prediction [14].

All the above points of weakness are overcome by the DML, as discussed above.

## 6. Conclusions and Future Perspectives

In the previous papers, the DML has shown its capabilities in the calculation of the specific heat of liquids, thermal conductivity, the modeling of diffusion, the relevance, and the calculation of the relaxation times involved in the liquid dynamics at the mesoscopic scale, the justification of the Cattaneo equation for heat propagation, the self-diffusion in pure liquids, etc. In the present manuscript, its application field has been extended to the modeling of thermodiffusion and of the Soret equilibrium, as well the Dufour effect. In particular, the DML provides a simple model not only for positive but also for negative values of $S_T$. Moreover, DML has been used to provide the first-ever theoretical interpretation of a new unexpected mechano-thermal effect discovered in liquids in shear geometry; at the same time, DML represents, to the author's knowledge, the first theory able to explain the conversion of mechanical energy into thermal energy, either in pure liquids or in liquid mixtures. Comparison with experimental results has been provided for all the cases considered.

The three major ingredients of the DML that make it new with respect to other liquid models are: (i) The dual character of liquids, supposedly composed by quasi-solid *liquid particles* and wave-packets, or *lattice particles*. (ii) The interaction between these two populations, in particular, its inelastic character and the time reversibility; the first allows modeling the mutual transfer of momentum other than energy between the *lattice* and the *liquid particles*, and the second makes this interaction as a good candidate for the elementary interaction at the base of the Onsager's time-reversibility that characterizes the NET. (iii) The so-called "tunnel effect" of the interactions between *lattice* and *liquid particles*, whose effects deeply influence the transient phases of a thermodynamic perturbation on the system.

Although there are still many aspects characterizing the liquid structure and their dynamics to be verified, a large field of applications for the DML represents a good test bed for such a model, also in view of further developments. The next big challenge will certainly be the modeling of viscosity, it will not be a surprise if many insights and similarities are found with the works of Eyring [104–106,122–124]. Recently [125], experiments performed in Newtonian fluids, such as water and glycerol, revealed for the first time the presence and long-lasting permanence of elasticity on a mesoscopic scale. The elastic response in such liquids persisted indeed for time laps of microseconds, hence four orders of magnitude longer than the typical intermolecular relaxation time. Such rubber-like elasticity and large strain response in fluid glycerol were interpreted by the authors as due to the relaxation processes of collective modes in metastable groups of molecules (those we have dubbed icebergs in the DML). Consequently, one may argue that the lifetime of such icebergs is equal to, if not larger than, the relaxation time of elastic propagation, as already hypothesized in

a previous paper [17]. The authors also point out that such modes require the existence of a transient state with solid-like long-range correlations, although different from the bulk state. This study has then revealed that liquid shear elasticity could be very underestimated on such a mesoscopic scale and high-frequency domain with respect to what is measured with the classical methods on a macroscopic scale. The same effect could affect also the viscosity value on the same scale, hence it is arguable that measuring viscosity at a mesoscopic scale in high frequency domain may reveal a response very different with respect to that typical of liquids. Once again, the DML is potentially able to provide the basis for the viscosity modeling in Newtonian fluids. This shall be the topic of future works.

Understanding how liquid parameters, as correlation lengths, sound velocity, thermal conductivity, etc., evolve from equilibrium to non-equilibrium conditions is an interesting topic that could be achieved by performing experiments in non-stationary temperature gradients. Also very interesting is to understand whether and how a temperature gradient affects the viscous coupling, and this could be understood by performing experiments by applying a temperature gradient to a stabilized isothermal liquid, ensuring that the average temperature remains unchanged, preventing the convection instability and performing light scattering experiments during the transient, exactly the situation that is normally avoided in all the experiments. An alternative way is to heat a small volume of liquid by hitting it with a focalized high-power laser beam.

One point that deserves deepening is certainly that of the evaluation of the parameter $m$, i.e., the collective DoF available at a given temperature and pressure for a *liquid particle.* In one of the previous papers [17], generic numerical results on dual models have already been commented on, although the calculation of the parameter $m$ necessitates to be supported by specific calculations. This and other simulations will be the core of future works currently in preparation.

The proposed interpretation of the mechano-thermal effect discovered by Noirez and co-workers stimulated the suggestion for a further experimental set-up that could shed more light on such phenomenon, namely: 1. It would be very interesting if it were possible to carry out a scattering experiment in the liquid cavity, although this would imply serious experimental difficulties; 2. To carry out the experiment by moving both plates in opposite directions (provided to have a suitable rheometer). If, on one side, this certainly increases the momentum flux transferred to the liquid, and hence to the phonons, on the other, it becomes extremely interesting and instructive to see "if" and "how" the temperature gradient is formed, and also verify the $L^{-3}$ rule. 3. To evaluate a possible effect due to the extension of the surface area of the two plates (see Equations (40)–(42)).

We want to close the paper with a further consideration of the capabilities of the DML, trying to answer the question of why an approach based on non-equilibrium statistical mechanics of a dual system succeeds whereas other molecular, traditional approaches fail. The gold example is the Boltzmann equation and its application to transport processes in dense fluids. Prigogine [126] highlighted that it relates the derivatives of the one-particle velocity distribution function with the collision operator. Such collisions are characterized by their instantaneity and localization at a point in the space. This simplification may be fruitful in gases but can fail when applied to liquids, where the duration of the interactions can be of the same order of magnitude as the interval between two successive interactions or of the relaxation times. The DML approach, although considering the interactions in a scheme that is simple and powerful at the same time, bypasses this weak point by means of the tunnel effect and considers the interactions as inelastic. The tunnel effect allows introducing the relaxation time(s) and moreover separates the interactions one from the other. The inelastic character ensures the capability of exchanging momentum other than energy between the two reservoirs, that of the *liquid particles* and the *lattice particles*. This last point will be the core of future works.

**Funding:** This research received no external funding.

**Data Availability Statement:** Not applicable.

**Acknowledgments:** The author wants to express his gratitude to the Reviewers who considerably enriched the manuscript and improved its quality.

**Conflicts of Interest:** The author declares no conflict of interests.

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
