# Peer review of "Thermo-Mechanical and Mechano-Thermal Effects in Liquids Explained by Means of the Dual Model of Liquids"

_2673-7264, doi:10.3390/thermo3040037_

Round 1
Reviewer 1 Report
Comments and Suggestions for Authors
Please look at the attachment

The quality of English Language is fine.
Reviewer 2 Report
Comments and Suggestions for Authors
See enclosed referee report

Reviewer 3 Report
Comments and Suggestions for Authors
SUMMARY:
This manuscript applies a previously proposed framework introduced by the author (see references [17-19] of this manuscript), called the Dual Model of Liquids (DML), to certain thermo-mechanical and mechano-thermal effects in liquids. The manuscript proposes expressions for the so-called Soret coefficient related to thermodiffusion and addresses a recently experimental mechanical-thermo effect in rotating liquids (see references [6-11] of this manuscript]. An attempt to compare the prediction of the proposed framework with experimental results is also performed.
COMMENTS:
The methodology and results presented in this article are interesting and could have potential. However, I believe the manuscript suffers from a weakness related to its length. I believe the manuscript's content is too large and full of redundancies. Instead of clarifying subtle points of the framework, I think it worsens the understanding of the suggested framework and its predictions.
RECOMMENDATION:
This manuscript presents a very interesting theoretical framework but needs to be improved in structure and length. I think more effort needs to be made to condense the message of this work, particularly how the predictions of the framework agree with experiments. After that, the potential of the proposed framework can be better appreciated by the reader. It is not necessary to repeat previously published details.
Comments on the Quality of English LanguageMinor English corrections are in order.
Reviewer 4 Report
Comments and Suggestions for Authors
Title: Thermo-Mechanical and Mechano-Thermal Effects in Liquids Explained by Means of the Dual Model of Liquids.
The author investigates theoretically the thermodiffusion, using the Dual Model of Liquids (DML). In the paper, previous research works on this topic are very well presented, setting the foundations for the analysis of the current work. Following are the remarks. In view of the remarks, I recommend for major revision of the manuscript.
· What is the novelty of the paper? The authors should mention more details in the introduction or conclusions.
· The comparison of both equations with experimental data is inadequate. Perhaps the author should mention more statistical and quantitative analysis with others works.
· It should be given more details about numerical solution of governing equations.
Round 2
Reviewer 1 Report
Comments and Suggestions for Authors
I thank the author who deeply revised the article, making it even more interesting. This is an excellent paper.
However certain details could be still improved:
- line 258, add « Inelastic » in front of Neutron Scattering for (INS)
- line 81 : please remove « disc rotation » and introduces « shear strain ».
- line 241: third law of thermodynamics
- line 747: I would prefer "two coaxial disk-like plates, one is fixed... the other one is mobile
- line 760: the sentence should be a bit improved. Indeed, at these low frequency and strain amplitudes, the energy stored cannot generate viscous heating following the classical empirical evaluation of the a Nahme number (also known as Brinkman number). Much faster velocities are needed for viscous liquids, closer to sound velocities.
- line 1113: "shear speed" or motion rather than "rotation" since the system did much much less than one revolution. Idem on 1114, 1124 and 1128, in agreement with the cited reference.
Reviewer 3 Report
Comments and Suggestions for Authors
I thank the author for making an effort to address my comments. I now believe the proposed theoretical framework described within this manuscript can be published in this journal.
Author Response
The author is grateful to the Reviewer for the suggestions given during the review process and for having accepted the manuscript in the final version.
Reviewer 4 Report
Comments and Suggestions for Authors
I would like to thank the author for clarifications and revisions. The manuscript is now accetable for publication.
Author Response

(The authors gave the same response as above.)
